# High-Dimensional Online Change Point Detection with Adaptive Thresholding and Interpretability

## Abstract

Change point detection (CPD) identifies abrupt and significant changes in sequential data, with applications in human activity recognition, financial markets, cybersecurity, manufacturing, and autonomous systems. While traditional methods often struggle with the computational demands of high-dimensional data, they also fail to provide explanations for detected change points, limiting their practical usability. This paper introduces a CPD framework that enhances both interpretability and scalability by leveraging the Sliced Wasserstein (SW) distance. Our contributions are fourfold: (1) we present a method to transform multivariate data into one-dimensional time series using the SW distance, enabling compatibility with existing CPD methods; (2) we derive theoretical insights, demonstrating that random slices of the SW distance follow a Gamma distribution, which facilitates statistical hypothesis testing for CPD; (3) we propose a novel self-adapting online CPD algorithm based on an adaptive threshold for a given significance level $q$; and (4) we propose a model-specific framework for generating contrastive explanations for annotated change points. We find that our method outperforms popular (online/offline) change point detection methods by reducing false positives by at least $48\%$ on average while also providing interpretable change points and maintaining competitive or superior detection performance, making it practical for deployment in high-stakes applications.

## 1 Introduction

Change point detection (CPD) is a fundamental problem in statistical analysis, focusing on identifying abrupt and significant changes in the underlying data-generating processes of sequential data. These changes can signal shifts in critical properties, such as distributions, relationships, or trends, making CPD pivotal in fields where timely detection of such shifts is crucial. Closely related to concept drift detection Gama et al. (2014); Harel et al. (2014); Lu et al. (2018), CPD encompasses scenarios of both abrupt and gradual changes, with a direct impact on the accuracy and reliability of machine learning models and deployed systems. However, existing CPD methods are insufficient in both scaling to high dimensions and providing meaningful explanations, which poses a significant gap addressed by our approach.

The significance of CPD becomes evident in its multitude of real-world applications. In *human activity recognition*, it can identify transitions between states, such as detecting when a person moves from walking to running Xia et al. (2020). In *financial markets*, CPD is essential for spotting regime shifts, such as the transition from a bull to a bear market, enabling traders and algorithms to adjust strategies Kim et al. (2022); Carvalho & Lopes (2007); Chen & Gupta (1997); Nystrup et al. (2016). In *cybersecurity*, CPD helps detect anomalies, such as cyberattacks or data breaches, by identifying abrupt deviations in network traffic Kurt et al. (2018); Polunchenko et al. (2012). Similarly, in *manufacturing quality control*, CPD can pinpoint defects or process anomalies to minimize waste and downtime. Furthermore, in *autonomous driving*, detecting changes in environmental conditions or sensor data ensures safe operation under dynamic conditions Ferguson et al. (2014); Galceran et al. (2017). These examples underscore the critical role of CPD in enhancing decision-making and ensuring the safety, efficiency, and reliability of systems across domains.

Despite its utility, CPD faces significant challenges when applied to high-dimensional data, where both scalability and explainability are becoming increasingly challenging. Traditional methods often rely on comparing probability distributions or distances between data segments to detect changes Aminikhanghahi & Cook (2017); Lu et al. (2018). While effective in lower-dimensional settings, these methods struggle with computational efficiency and scalability in higher-dimensional spaces. For instance, the exact computation of the Wasserstein distance for multivariate data scales as $\mathcal{O}(n^3 \log(n))$, making it impractical for large datasets. Similarly, the computation of $U$- and $V$-statistics for the Maximum Mean Discrepancy (MMD) also scales quadratically in time. Alongside the computational aspects, most CPD methods fail to provide interpretable change points, narrowing down the root cause of the drifts.

To address the lack of explainable change point detection tailored for high-dimensional data, the Sliced Wasserstein (SW) distance Bonneel et al. (2015) offers a promising alternative. Instead of computing a high-dimensional optimal transport directly, we can repeatedly project onto a single dimension, where Wasserstein distance has a closed form, and then average the results. By leveraging the closed-form expression of the Wasserstein distance for one-dimensional distributions, the SW distance reduces the computational complexity to $\mathcal{O}(n \log(n))$ by averaging over the Wasserstein distances of random one-dimensional projections. Additionally, by leveraging the geometric properties of the random projections, we can provide contrastive explanations for detected change points.

In this work, we bridge this gap by introducing a novel CPD framework that leverages the Sliced Wasserstein distance. Our contributions are as follows:

1. **A Self-Adapting Online CPD Algorithm with Adaptive Thresholding (3.2).** We propose a new self-adapting online CPD algorithm that dynamically adjusts its threshold based on a given significance level $\alpha$. This enables robust and adaptive detection of change points in streaming high-dimensional data without manual tuning.

2. **Theoretical Insight: SW Distance Slices Follow a Gamma Distribution (3).** We derive a novel theoretical result showing that random slices of the SW distance follow a Gamma distribution. This allows for a principled statistical hypothesis testing framework, enabling more rigorous and interpretable change detection.

3. **Contrastive Explanations for Change Points Using Geometric Properties of SW Distance (3.1).** We develop a novel, model-specific framework for generating contrastive explanations of detected change points. By leveraging the geometric properties of random projections, we provide fine-grained insights into which features contribute most to distributional shifts, enhancing interpretability.

4. **Competitive Performance with Interpretability (4.2)** Our approach achieves competitive or superior performance compared to leading online and offline CPD methods across multiple real-world datasets while providing interpretable change points, making it practical for deployment in high-stakes applications such as finance, cybersecurity, and autonomous systems.

## 2 RELATED WORK

**Online change point detection.** Change point detection can be grouped into parametric and nonparametric methods Truong et al. (2020). Parametric methods assume that the data is drawn from some parametric family of probability distributions. Nonparametric approaches do not impose distributional assumptions. One of the most prominently known parametric approaches is the cumulative sum (CUSUM) method Page (1954). Over the last years, several extensions of CUSUM were introduced Alippi & Roveri (2006); Romano et al. (2023); Yu et al. (2023). Another popular parametric branch of change point detection are Bayesian methods including Fearnhead & Liu (2007); Knoblauch et al. (2018). Nonparametric methods are often based on test statistics derived by distances, including Euclidean distances Matteson & James (2014); Madrid Padilla et al. (2019) or divergence measures e.g. MMD Gretton et al. (2012); Harchaoui et al. (2013); Li et al. (2019) or test-statistics based on density-ratio estimation Sugiyama et al. (2008); Kanamori et al. (2009); Yamada et al. (2013); Liu et al. (2013b). More recently, deep generative models Chang et al. (2019); De Ryck et al. (2021) and density-ratio estimation based on deep neuronal networks Hushchyn et al. (2020); Hushchyn & Ustyuzhanin (2021) were also used for sequential change point detection.

**Optimal transport based change detection.** Over the past few years, optimal transport has become a popular choice for comparing two distributions. Naturally, optimal transport-based metrics, such as the Wasserstein distance or Sliced Wasserstein distance, can also be applied for sequential change point detection. This includes Cheng et al. (2020a), which proposes a change point detection framework computing the Wasserstein distance between a sliding window relying on a fixed threshold to detect changes. Similar approaches were introduced in Faber et al. (2021; 2022). In Cheng et al. (2020b), this framework was refined using a matched filter test statistic. Furthermore, one of the proposed test statistics is the Sliced Wasserstein distance, which is combined with a fixed threshold. Our work differs by introducing an adaptive threshold and primarily investigating the Sliced Wasserstein distance as a tool for interpretability.

**Interpretability through random projections.** The motivation behind utilizing random projection is the lower computational cost for the Wasserstein distance. In Wang et al. (2021), a projected Wasserstein distance was introduced, which finds a k-dimensional subspace through linear projections and calculates the Wasserstein distance in the lower-dimensional space. Analogously, in Wang et al. (2022), the kernel projected Wasserstein distance was motivated as a non-linear alternative to Wang et al. (2021). Both approaches reduce the computational complexity and facilitate interpretability in a two-sample test. Our proposed framework goes beyond a single iteration to find a specific projection direction, maximizing the Wasserstein distance between projected samples. We propose an iterative approach to identify the most discriminative feature, leading to a more comprehensive and detailed explanation of the underlying drift.

## 3 PROBLEM SETUP

The general problem of CPD involves determining abrupt changes in a time series. We denote the time series $\mathcal{D} = \{x_t \in \mathbb{R}^d : t \in [T]\}$ with $[T] = \{1, 2, \ldots, T\}$ and assume that the time series follows some unknown underlying distribution $\mathbb{P}$. The goal is to identify all timestamps $t_* \in [T]$ where the underlying distribution changes from $\mathbb{P}$ to $\mathbb{Q}$, such that

$$t \leq t_* : x_t \sim \mathbb{P}$$
$$t > t_* : x_t \sim \mathbb{Q}.$$

Many CPD methods rely on a windowing approach and split the observations into a reference window $X_t^r = \{x_{t-k}, \ldots, x_{t-1}\}$ and current/test window $X_t^c = \{x_t, \ldots, x_{t+k}\}$ with $k$ observations and deploy a hypothesis test or calculate a distance between the two windows and compare it against a threshold at each timestamp.

Consider $\mathbb{P}, \mathbb{Q}$ to be two probability distributions with $p$ finite moments. The Wasserstein distance, denoted as, $W_p^p(\mathbb{P}, \mathbb{Q})$ has a closed expression for univariate distributions,

$$W_p^p(\mathbb{P}, \mathbb{Q}) = \int_0^1 |F^{-1}(u) - G^{-1}(u)|^p \mathrm{d}u \tag{1}$$

where $F^{-1}, G^{-1}$ are the inverse CDF of $\mathbb{P}$ and $\mathbb{Q}$ respectively. The sliced Wasserstein distance (SW) exploits this closed expression by averaging over the Wasserstein distance between infinitely many random one-dimensional projections of $\mathbb{P}$ and $\mathbb{Q}$. In particular, for any direction $\theta \in \mathbb{S}^{d-1}$, we define the projection of $x \in \mathbb{R}^d$ as $T^\theta(x) = \langle x, \theta \rangle$ and denote the projected distribution with $\mathbb{P}^\theta = T_\#^\theta \mathbb{P}$, where $\#$ is the push-forward operator, defined as $T_\# \mathbb{P}(A) = \mathbb{P}(T^{-1}(A))$ for any Borel set $A \in \mathbb{R}^d$. Let us denote $\lambda$ the uniform measure on $\mathbb{S}^{d-1} = \{\theta \in \mathbb{R}^d : ||\theta||_2 = 1\}$, then the $p$ Sliced Wasserstein distance between $\mathbb{P}$ and $\mathbb{Q}$ is defined as

$$SW_p^p(\mathbb{P}, \mathbb{Q}) = \int_{\mathbb{S}^{d-1}} W_p^p(\mathbb{P}^\theta, \mathbb{Q}^\theta) \mathrm{d}\lambda(\theta). \tag{2}$$

In practice, the computation of the SW boils down to a Monte Carlo approximation by uniformly sampling projection parameters $\{\theta_l\}_{l=1}^L$ on $\mathbb{S}^{d-1}$ and average over the one-dimensional Wasserstein distances obtained. Let us denote the slice $w_p^p : \theta \mapsto W_p^p(\mathbb{P}^\theta, \mathbb{Q}^\theta)$ as a function mapping a projection direction to the p Wasserstein distance. Then, we have the Monte Carlo approximation, $\widehat{SW_p^p}(\mathbb{P}, \mathbb{Q}) = L^{-1} \sum_{l=1}^L w_p^p(\theta_l)$ accordingly. The accuracy of this estimator heavily relies on the variance of $w_p^p$ Nietert et al. (2022). Based on the following result, we derive the adaptive threshold, which is based on the MoM estimated parameters of a Gamma distribution.

**Theorem 3.1.** *Let $\mathbb{P}, \mathbb{Q}$ denote two probability distributions on $\mathbb{R}^d$ with finite $p$'th moments then $w_2^2(\theta)[\mathbb{P}^\theta, \mathbb{Q}^\theta] \sim \Gamma$ as $d \to \infty$*

The following Proposition allows us to consider the uncertainty of the Method of Moments (MoM) estimates based on the observed samples for the adaptive threshold.

**Proposition 3.2.** *Suppose some i.i.d. samples $X_n = (x_1, \ldots, x_n)$ with $x_i \sim \Gamma(\alpha, \beta)$ for $i = 1, \ldots, n$ with sample mean $\overline{X}_n = \frac{1}{n} \sum_{i=1}^{n} x_i$ and sample variance $S_n^2 = \frac{1}{n-1} \sum_{i=1}^{n} (x_i - \overline{X}_n)^2$. Then, the two-tailed confidence intervals for confidence level q of the Method of Moments (MoM) estimates $\widehat{\alpha}, \widehat{\beta}$ are*

$$C_p(\widehat{\alpha}) = \left[ \widehat{\alpha} - z_{\frac{q}{2}} \cdot \sqrt{\text{Var}(\widehat{\alpha})}, \widehat{\alpha} + z_{\frac{q}{2}} \cdot \sqrt{\text{Var}(\widehat{\alpha})} \right]$$

$$C_p(\widehat{\beta}) = \left[ \widehat{\beta} - z_{\frac{q}{2}} \cdot \sqrt{\text{Var}(\widehat{\beta})}, \widehat{\beta} + z_{\frac{q}{2}} \cdot \sqrt{\text{Var}(\widehat{\beta})} \right] \tag{3}$$

*where $z_{\frac{q}{2}}$ is the z-value of a standard normal distribution for confidence level q, and*

$$\text{Var}(\hat{\alpha}) \approx \frac{6\alpha^2}{n}, \quad \text{Var}(\hat{\beta}) \approx \frac{\beta^2 + 2\alpha\beta^2}{n\alpha}$$

### 3.1 EXPLAINABILITY

We denote the collection of random slices between $\widehat{\mathbb{P}}_n, \widehat{\mathbb{Q}}_n$ with $S_L(\widehat{\mathbb{P}}_n, \widehat{\mathbb{Q}}_n) = \{w_2^2(\theta_l)\}_{l=1}^{L}$, the empirical mean of $S_L$ is the Monte Carlo approximation of $\text{SW}_2^2(\widehat{\mathbb{P}}_n, \widehat{\mathbb{Q}}_n)$. We can interpret $w_2^2(\theta_l)$ as the loss for projection direction $\theta_l$. In this case, the loss quantifies the Wasserstein distance of the corresponding projection. We can use the linkage between projection direction and Wasserstein loss $w_2^2$ to derive a feature importance. We propose to average over the absolute projections parameters corresponding to the slices above the $q$-quantile of $S_L$. The procedure is illustrated in Algorithm 1.

We use a hierarchical approach to obtain contrastive explanations for change points. We start to identify the feature dimension achieving the highest feature contribution according to algorithm 1. Then, we eliminate the dissimilarity for this feature dimension by replacing the values with the mean of the same feature of the reference set, and validate the feature removal step by calculating random projections $S_L$ between the updated sample sets. This step indicates whether the reduced sample sets still contain drifted feature dimensions since under

---

**Algorithm 1** Calculate Feature Contribution
**Input:** Slices $\mathbf{S_L}$, Projection parameters $\boldsymbol{\theta}$, Wasserstein order: $\mathbf{p}$, Quantile level: $\mathbf{q}$

1: $S_L^{\rightarrow} = [w_p(\theta_{\pi(1)}), w_p(\theta_{\pi(2)}), \ldots, w_p(\theta_{\pi(L)})]$  ▷ Sort $S_L$ in ascending order
2: $\theta_{1:L}^{\rightarrow} = [\theta_{\pi(1)}, \theta_{\pi(2)}, \ldots, \theta_{\pi(L)}]$  ▷ Sorted $\theta$ according to $S_L$
3: $i_q \leftarrow \lceil qL \rceil$  ▷ Compute quantile index
4: $I_s = \frac{1}{L-i_q} \sum_{i=i_q}^{L} |\theta_{\pi(i)}|$
5: **Return** $I_s$

---

$H_0$, both samples arise from the same underlying process, and the SW between the empirical distributions approaches 0. We propose a stopping criterion based on the norm of the mean differences which is upper bounded by some constant given in terms of $d$, $N$, and the covariance matrix. We derive the stopping criterion in AppendixB.4. Our proposed model-specific explanation procedure is illustrated in Algorithm 2.

### 3.2 PROPOSED DETECTION METHOD

The main observation is that $S_L(\mathbb{P}, \mathbb{Q})$ follows a Gamma distribution with $\text{SW}_p^p(\mathbb{P}, \mathbb{Q}) = \mathbb{E}[S_L]$. We process the data in an online manner with a sliding window of $w$ observations and write

$$\mathcal{D}_t^w = \{\underbrace{x_{t-w}, \ldots, x_{t-w+\lfloor \frac{w}{2} \rfloor}}_{\mathbb{P}}, \underbrace{x_{t-w+\lfloor \frac{w}{2} \rfloor+1}, \ldots, x_t}_{\mathbb{Q}}\},$$

for $t \geq w$ which means the change point detection procedure is initiated after observing $w$ data samples. Furthermore, we denote the probability distribution of the first half of the sliding window with $\mathbb{P} = \lfloor \frac{w}{2} \rfloor^{-1} \sum_{i=0}^{\lfloor \frac{w}{2} \rfloor} \delta_{x_{t-w+i}}$ and the second half with $\mathbb{Q} = (\lfloor \frac{w}{2} \rfloor + 1)^{-1} \sum_{i=0}^{\lfloor \frac{w}{2} \rfloor+1} \delta_{x_{t-i}}$. After

---

**Algorithm 2** Hierarchical validated explanations

**Input:** Data: $\mathbf{X}, \mathbf{Y}$, Wasserstein order: $\mathbf{p}$, Quantile level: $\mathbf{q}$, Number of projections: $\mathbf{L}$

---

1: $\text{cl} \leftarrow [1, \ldots, d]$ ▷ Track which features are left
2: $\text{cr} \leftarrow \emptyset$ ▷ Removed features
3: $C \leftarrow \sqrt{\frac{2}{N} \text{tr}(\Sigma_X)}$
4: **while** $||D|| \geq C$ **and** $|\text{cl}| > 0$ **do**
5:      Calculate random projections $\mathbf{S}_L$
6:      Calculate Feature Contributions $I_s$ ▷ Algorithm 1
7:      $i_* \leftarrow \arg\max I_s$ ▷ Find feature with highest contribution
8:      $\text{cr} \leftarrow \text{add}(i_*, \text{cr})$
9:      $\mathbf{Y}[:, i_*] \leftarrow \mathbb{E}[\mathbf{X}[:, i_*]]$ ▷ Update feature
10:      $D \leftarrow \frac{1}{N} \sum_{i=1}^{N} X_i - \frac{1}{N} \sum_{i=1}^{N} Y_i$
11: **end while**
12: **Return** cr

---

observing $k$ samples, we calculate $S_L(\mathbb{P}, \mathbb{Q}) = S_L(\mathcal{D}_t^w)$ and initially fit the data to a Gamma distribution. Using the Method of Moments (MoM), we obtain a parameter estimation with

$$\widehat{\alpha} = \frac{\overline{S_L}^2}{\mathbb{V}(S_L)}, \quad \widehat{\beta} = \frac{\overline{S_L}}{\mathbb{V}(S_L)} \tag{4}$$

where $\overline{S_L}$ denotes the sample mean of $S_L$, implying $\widehat{\text{SW}}(\mathbb{P}, \mathbb{Q}) = \frac{\widehat{\alpha}}{\widehat{\beta}}$, and $\mathbb{V}(S_L)$ denotes the sample variance of $S_L$. Proposition 3.2 enables us to calibrate confidence intervals for MoM estimated $\hat{\alpha}_t, \hat{\beta}_t$ for each time step $t$. In the following, we propose an adaptive online detection method (SWCPD) that monitors the cumulative Sliced Wasserstein distances against a dynamic threshold. At each time step $t$, the procedure consists of the following steps:

(1) UPDATE CUMULATIVE SUM: We compute the expected value of the test statistic

$$C_t = C_{t-1} + \mathbb{E}[S_L(\mathcal{D}_t^w)],$$

(2) PROPAGATE MoM ESTIMATES: In a sliding window, there are dependencies between successive data windows. We smooth past MoM estimates using a moving average over the most recent $m = \min\{K_{max}, t\}$ steps with

$$\mathbb{E}[\hat{\alpha}_{t+1}|C_t] = \frac{1}{m} \sum_{i=t-m}^{t} \hat{\alpha}_i \quad \mathbb{E}[\hat{\beta}_{t+1}|C_t] = \frac{1}{m} \sum_{i=t-m}^{t} \hat{\beta}_i.$$

Despite temporal correlations, the i.i.d. nature of the random projections ensures the validity of our statistical bounds. (3) BOUND CUMULATIVE SUM: We use the smoothed MoM estimates to bound the next step in the cumulative sum via the quantile of the corresponding Gamma distribution:

$$\mathbb{E}[C_{t+1}|C_t] = C_t + \mathbb{E}\left[\frac{\hat{\alpha}_{t+1}}{\hat{\beta}_{t+1}}\Big|C_t\right] \leq C_t + \kappa(q)$$

where $\kappa(q)$ denotes the $q$-quantile of $\Gamma(\hat{\alpha}_{t+1}, \hat{\beta}_{t+1})$.

(4) VALIDATE DEVIATIONS: After observing $\mathcal{D}_{t+1}^w$, we update $C_{t+1}$, and compare it against the upper bound. If it exceeds the bound, a change point is detected. The MoM estimates are then updated using the new data.

## 4 EXPERIMENTS

We first evaluate the alignment of feature explanations obtained with the SW distance and Algorithm 2 to SoTA feature explanation methods. We demonstrate that Algorithm 2 leads to informative insights that enable contrastive explanations for change detection. In the second part of this section, we show the feasibility of our method against various popular offline and online change point detection methods, achieving comparable or better results.

Table 1: Mean alignment (eq. (6)) of SWD explanations with IG, GS, and DL explanations for dimensions $d = 10, 20$ and various number of drifted components $k = 1, 3, 7, 9$ over 5 different runs.

| | $d = 10$ | | | $d = 20$ | | |
|---|---|---|---|---|---|---|
| | IG | GS | DL | IG | GS | DL |
| $k = 1$ | $0.959 \pm 0.048$ | $0.962 \pm 0.045$ | $0.965 \pm 0.041$ | $0.994 \pm 0.001$ | $0.994 \pm 0.001$ | $0.994 \pm 0.002$ |
| $k = 3$ | $0.940 \pm 0.048$ | $0.940 \pm 0.046$ | $0.939 \pm 0.040$ | $0.950 \pm 0.039$ | $0.950 \pm 0.040$ | $0.947 \pm 0.042$ |
| $k = 7$ | $0.900 \pm 0.027$ | $0.902 \pm 0.028$ | $0.900 \pm 0.043$ | $0.924 \pm 0.022$ | $0.923 \pm 0.020$ | $0.923 \pm 0.024$ |
| $k = 9$ | $0.885 \pm 0.031$ | $0.885 \pm 0.030$ | $0.855 \pm 0.027$ | $0.924 \pm 0.022$ | $0.924 \pm 0.020$ | $0.936 \pm 0.015$ |

## 4.1 EXPLAINABILITY

We evaluate feature explanations using the SW distance (SWD) and compare it to SoTA feature explanations obtained with Integrated Gradients (IG) Sundararajan et al. (2017), Gradient Shap (GS) Lundberg & Lee (2017), and DeepLIFT (DL) Shrikumar et al. (2017) for synthetic data and real-world data.

**Synthetic Data.** We generate data $X_{1:N} \sim \mathcal{N}(\mu_d, \Sigma_d)$ for $N = 5000$ and $d = 10, 20$, with mean $\mu_d$ and covariance $\Sigma_d$. Each component of $\mu_d^i$ follows a normal distribution and is sampled independently.

We randomly select $k \leq d$ indices in $\mu_d$ and sample an individual severity $\epsilon_i \sim \mathcal{N}(2, 1)$ for each selected index, which is added to the mean prior to the drift $\tilde{\mu} = \mu + \epsilon$. This ensures that some feature dimensions are more important for the total drift and should show a higher contribution to the explanation scores. We generate data after the drift $\tilde{X}_{1:N} \sim \mathcal{N}(\tilde{\mu}_d, \Sigma_d)$, throughout the experiments, we vary the number of drifted components $k = 1, 3, 7, 9$ and set $\Sigma_d = \mathbb{I}_d$. For a binary classification of samples before and after the drift, we train a simple fully connected neural network with three hidden layers with $128, 64,$ and $32$ units, respectively. We use IG, GS, and DL to calculate feature attributions $\phi(X), \phi(\tilde{X})$ for data before and after the drift occurred. For SWD, we follow Algorithm 2 to

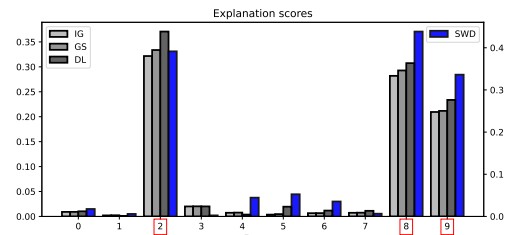

Figure 1: Explanation scores for each feature obtained with IG, GS, DL, and SWD (higher score indicates a higher importance). Red boxes indicate ground truths (drifts).

assign explanation vector $e_{\text{SWD}}$. To quantify how severe the differences in the attribution scores for IG, GS, and DL are, we assign some explanation scores by calculating the absolute differences between both attributions

$$e := |\phi(X) - \phi(\tilde{X})|. \tag{5}$$

In Figure 1, we visualize the explanation scores for each feature for some data with $d = 10$ and $k = 3$. The red boxes indicate the drifted features and mark the ground truths. We see that all reference methods show similar explanation scores, and SWD-based explanations have a strong alignment with the reference methods. We use the cosine similarity to quantify the alignment between SWD and the reference explanation vectors,

$$s(e, e_{\text{SWD}}) = \frac{\langle e, e_{\text{SWD}} \rangle}{||e||_2 ||e_{\text{SWD}}||_2}. \tag{6}$$

We investigate the alignment for different scenarios by varying $d = 10, 20$ and $k = 1, 3, 7, 9$. For each parameter pair, we simulate data and calculate alignment between SWD explanation scores and IG, GS, and DL for five different runs. In Table 1, we report the average alignment between SWD explanations and explanations obtained by IG, GS, and DL after the first iteration of Algorithm 2.

**Real World Data.** We employ a Vision Transformer (ViT) model Dosovitskiy et al. (2021) for image classification on the MNIST LeCun et al. (2010) dataset. Details on the model architecture can be found in section B.1.1. We simulate a streaming behavior of samples from a particular class, which then abruptly changes to another class. The feature attributions before and after the drift will differ w.r.t. to the underlying feature characteristics of each class. We split the test dataset for each class

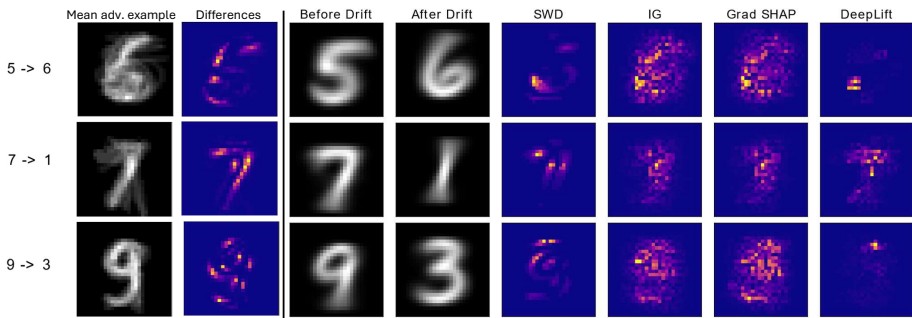

Figure 2: Shows the average adv. example and its corresponding differences for three different drifts (left). On the right-hand side, we see the average example of each class before and after the drift alongside the highlighted feature attributions with SWD, IG, GS, and DL.

and calculated the feature attribution respectively. The average feature attribution per class shows the most important features for a given concept, e.g., number 7 has distinct characteristics (edges, curvature) to number 0. However, the general representation of number 1 should be similar to 7 on a feature level, such that the classification model indicates a substantial overlap in the feature attributions. We calculate the absolute differences of the average feature attributions for two classes using IG, GS, and DL, which we use as a qualitative measure to explain the drift. We modify the projection procedure in Algorithm 2 by using the unit vectors to obtain a pixelwise importance and terminate after 250 iterations. We found that for two distinct digits, there are 245.08 pixels on average, which show an absolute deviation above 0.1. Changes below this are generally indistinguishable, such that this reduced set captures the most important pixels which are a valid representation of the original class, therefore 250 is a conservative qualitative stopping criterion. Figure 2 shows the results for three challenging drifts. IG and GS show similar results, which is plausible since GS computes expected gradients and can be seen as an extension of IG. Sparsity is especially important for adversarial attacks, which aim to alter the model output with minimal perturbations of the inputs. The fast gradient sign method (FGSM) Goodfellow et al. (2014) is a prominent adversarial attack method that alters the input by the sign of the gradient of the loss function w.r.t the input to fool a model making incorrect predictions. We simulated adversarial attacks on the ViT model using FGSM with $\epsilon = 5 \times 10^{-4}$ and compared the average adversarial example to the average non-adversarial example, which can be seen in Figure 2. This illustrates which features are likely to be liable under attacks, thus principal to the model, which should also be reflected in the feature attributions.

### 4.2 CHANGE POINT DETECTION

In this part, we evaluate our proposed method on a synthetic dataset and four real-world datasets, namely MNIST, Human Activity Recognition (HAR) Anguita et al. (2013), Human Activity Segmentation Challenge (HASC) Ermshaus et al. (2023a), and Occupancy Candanedo & Feldheim (2016). While MNIST is challenging in the number of dimensions, the sensor data from HAR and HASC combines drifts in variance and means. We report Area under Curve (AUC) scores, segmentation covering scores, average detection delay, and the average number of false positives. For a detailed description and motivation for the used metrics, we refer the reader to Van den Burg & Williams (2020) and Ermshaus et al. (2023b). We compare our method against five popular change point detection methods (BOCPD Adams & MacKay (2007), e-divisive Matteson & James (2014), KCP Arlot et al. (2019), OT-CPD Cheng et al. (2020a), RuLIFS Liu et al. (2013a)), one time series segmentation method (ClaSP Ermshaus et al. (2023b)), and two deep learning based methods (ONNR,ONNC from Hushchyn et al. (2020); Hushchyn & Ustyuzhanin (2021) here called DeepRuLIFS, DeepCLF). Generally, an appropriate hyperparamter choice includes $w$ smaller than the average segment length, $K_{\max}$ the same size as $w$ or smaller fractions for a more adaptive threshold with a smaller autoregressive lag, $p = 2, 4$, sufficiently large $L > 500$, and $q < 0.15$ for a robust detection threshold. In the following, we briefly describe the datasets on which we conducted experiments and highlight subsequent results.

**Synthetic Data:** We construct a data stream of $d = 50$ exponential distributions $x_i \sim \text{Exp}(\lambda) + c_i$, where $c_i$ is randomly sampled within $(-3, 3)$ for $i = 1, \ldots, d$. We simulate 3 segments, where each

Table 2: Shows average AUC scores with standard deviation, and average number of false positives and detection delay with min-max values for synthetic data

| | Exponential | | | | | | Mixture | | | | |
|---|---|---|---|---|---|---|---|---|---|---|---|
| | AUC (↑) | | FP (↓) | | DD (↓) | | AUC (↑) | | FP (↓) | | DD (↓) |
| $\lambda$ | $\tau = 10$ | $\tau = 20$ | $\tau = 10$ | $\tau = 20$ | | $\sigma / \lambda$ | $\tau = 10$ | $\tau = 20$ | $\tau = 10$ | $\tau = 20$ | |
| 0.5 | $0.6 \pm 0.13$ | $0.93 \pm 0.13$ | 1.2 (1; 2) | 0.2 (0; 1) | 14.8 (11; 18.5) | 0.25 | $1.0 \pm 0.0$ | $1.0 \pm 0.0$ | 0 (0; 0) | 0.0 (0; 0) | 5.6 (3.5; 7.5) |
| 0.1 | $0.47 \pm 0.1$ | $0.55 \pm 0.17$ | 0.8 (0; 1) | 0.6 (0; 1) | 16.6 (0; 22) | 0.5 | $0.53 \pm 0.16$ | $0.87 \pm 0.16$ | 1.4 (1; 2) | 0.4 (0; 1) | 14.9 (10.5; 20.5) |

segment consists of 500 samples. We randomly select a total of 3 features for which we inject a drift by offsetting the mean $c_i$ randomly sampled within $(-3, 3)$ for each drifted feature. Additionally, we generated a mixture distribution consisting of 20 Exponential distributions and 30 Gaussian distributions. In Section C.2.1, we provide a detailed description of the sampling procedure. For all experiments on synthetic data, we set the window length $w = 50$, the lookback window for the estimation of shape- and rate parameters $K_{\max} = 50$, $p = 2$, and $L = 5000$. Table 2 shows the average AUC scores, number of false positives, and detection delay for Exponential- and mixture distributions for different distributional parameters $\lambda, \sigma$, and different detection thresholds $\tau$ in the calculation of AUC scores, false positives.

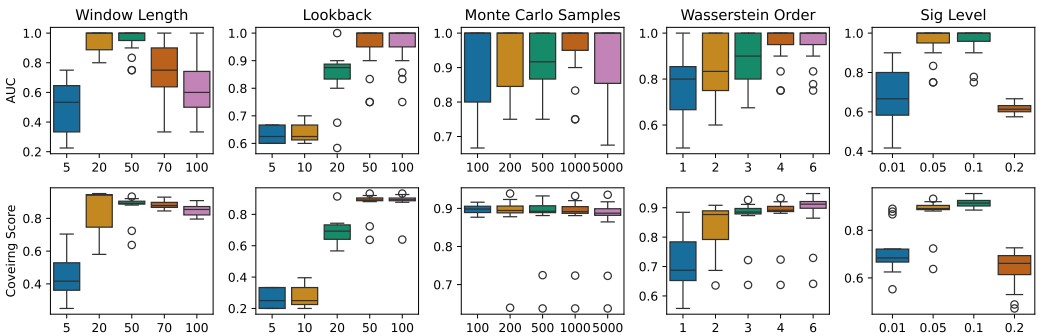

Figure 3: Boxplots of AUC and Covering scores for each parameter variation while keeping the other parameters fixed.

**Faithfulness:** Additionally, we investigate the faithfulness of *discriminative features* derived using Algorithm 2. For this matter, we simulate a $50/50$ mixture distribution of Gaussian and Exponential random variables with $d = 50$ with 500 observations. We randomly select 10 features for which we inject a mean shift at $t = 250$ with a magnitude uniformly sampled in $[-\delta, \delta]$. We let our method identify the 10 most discriminative features and mask the time series by removing the identified features. We use an independent oracle (KCP) with an AUC and covering score of 1.0 on the original data, and evaluate it on the masked data. We report the True Positive change $\Delta_{\text{TP}} = \text{TP}_{\text{clean}} - \text{TP}_{\text{masked}}$ and covering change $\Delta_{\text{Cov}} = \text{Cov}_{\text{clean}} - \text{Cov}_{\text{masked}}$. Since, $\text{TP}_{\text{clean}} = 1.0$, the desired $\Delta_{\text{TP}} = 1.0$ which indicates that without the discriminative features, the oracle no longer detects any change point. Thus, the desired covering change is $\Delta_{\text{Cov}} = 0.5$ as no segmentation leads to Cov$= 0.5$. Additionally, we calculate the discriminative accuracy as the fraction of identified discriminative features and ground truth discriminative features.

Table 3: Shows the average discriminative accuracy of Algorithm 2 and the influence on the detection ability measured by the change of true positives and covering score.

| $\delta$ | 0.2 | 0.3 | 0.5 | 0.7 | 1.0 | 2.0 |
|---|---|---|---|---|---|---|
| Acc | $0.68 \pm 0.11$ | $0.77 \pm 0.08$ | $0.87 \pm 0.08$ | $0.90 \pm 0.08$ | $0.90 \pm 0.08$ | $0.95 \pm 0.08$ |
| $\Delta_{\text{TP}}$ | $1.0 \pm 0.0$ | $1.0 \pm 0.0$ | $1.0 \pm 0.0$ | $1.0 \pm 0.0$ | $1.0 \pm 0.0$ | $1.0 \pm 0.0$ |
| $\Delta_{\text{Cov}}$ | $0.49 \pm 0.01$ | $0.49 \pm 0.01$ | $0.50 \pm 0.0$ | $0.50 \pm 0.0$ | $0.50 \pm 0.0$ | $0.50 \pm 0.0$ |

**MNIST:** In order to mimic a streaming behavior, we uniformly sample an initial class (without replacement) and select $K$ instances from the current class. We repeat this procedure and annotate the samples to introduce abrupt changes. Within the scope of the experiments for this paper, we generated 5 distinct data sequences with $2, 3$, and $4$ change points, where each class has 200 samples. In our experiments, SWCPD is able to deliver competitive AUC scores while delivering minimal false positives on average. Additionally, we conducted an ablation study to investigate the influence of each parameter on the AUC and Covering score. We observe that SWCPD's performance is mostly liable to proper specification of significance level and window length, see Figure 3. We report detailed results in Section C.1. For results in Table 4, we set $\tau = 20$, $w = 50$, $K_{\max} = 25$, $L = 5000$, $p = 4$, and $q = 0.1$.

**HASC & HAR:** Datasets consists of distinct multimodal multivariate time series monitoring human motion during various daily activities denoted as HASC and HAR. HASC data was collected as part of the Human Activity Segmentation Challenge Ermshaus et al. (2023a) using built-in smartphone sensors. In total, the dataset has 250 time series consisting of 12 different measurements sampled at 50 Hz, where the ground truth change points were independently annotated using video and sensor data. We selected 25 instances covering indoor and outdoor activities for various numbers of segments, ranging from 1 to 6. We specifically considered instances with a single segment to assess each method's robustness to false positives. We refer to Ermshaus et al. (2023a) for a thorough description of the data and cover some insights on the selected data in Section C.2.3. We set $w = 500$, $K_{\max} = 20$, $L = 500$, $p = 2$, and $q = 0.05$. We used a margin of 100, which corresponds to a maximum tolerated delay of two seconds in the calculation of precision and recall, and average number of false positives for Table 4. HAR Anguita et al. (2013) was collected from 30 volunteers who performed six daily activities (walking, sitting, etc.) while wearing a smartphone on their waist to record $3-$axis acceleration and angular velocity at 50 Hz using embedded sensors. Naturally, the change points are given when an activity changes. In total, there are 10.299 observation of $d = 561$ features. We set $\tau = 10$, $w = K_{\max} = 20$, $L = 5000$, $p = 2$, and $q = 0.075$.

Table 4: Shows the average AUC & Covering scores, average detection delay (DD), and false positives (FP) together with the standard deviation of SWCPD and comparison methods over real-world datasets. **Bold** numbers indicate best performance; underlined values are statistically equal to best results [1].

| Dataset | | e-divisive | KCP | BOCPD | CLasP | RuLSIF | DeepRuLSIF | DeepCLF | OT-CPD | SWCPD |
|---|---|---|---|---|---|---|---|---|---|---|
| | | | | | | | | | | **Method** |
| Occupancy | AUC (↑) | $0.34 \pm 0.0$ | $0.52 \pm 0.0$ | $0.57 \pm 0.0$ | $\underline{0.58} \pm 0.0$ | $0.38 \pm 0.0$ | $0.44 \pm 0.0$ | $0.40 \pm 0.0$ | $0.40 \pm 0.0$ | $\mathbf{0.59} \pm 0.0$ |
| | COV (↑) | $0.64 \pm 0.0$ | $0.64 \pm 0.0$ | $0.73 \pm 0.0$ | $0.19 \pm 0.0$ | $0.79 \pm 0.0$ | $0.78 \pm 0.0$ | $0.76 \pm 0.0$ | $0.73 \pm 0.0$ | $\mathbf{0.81} \pm 0.0$ |
| | DD (↓) | $\underline{53}\,(-;-)$ | $77\,(-;-)$ | $105\,(-;-)$ | $-\,(-;-)$ | $85\,(-;-)$ | $102\,(-;-)$ | $98\,(-;-)$ | $129\,(-;-)$ | $\mathbf{52}\,(-;-)$ |
| | FP (↓) | $12\,(-;-)$ | $11\,(-;-)$ | $11\,(-;-)$ | $-\,(-;-)$ | $8\,(-;-)$ | $8\,(-;-)$ | $7\,(-;-)$ | $11\,(-;-)$ | $\mathbf{4}\,(-;-)$ |
| MNIST | AUC (↑) | $\underline{0.96} \pm 0.05$ | $0.91 \pm 0.06$ | $0.69 \pm 0.15$ | $0.63 \pm 0.03$ | $0.63 \pm 0.03$ | $0.91 \pm 0.17$ | $0.93 \pm 0.1$ | $0.95 \pm 0.05$ | $\mathbf{0.97} \pm 0.07$ |
| | COV (↑) | $\underline{0.95} \pm 0.05$ | $0.93 \pm 0.05$ | $0.78 \pm 0.11$ | $0.26 \pm 0.06$ | $0.26 \pm 0.05$ | $0.92 \pm 0.04$ | $0.94 \pm 0.02$ | $\mathbf{0.96} \pm 0.10$ | $0.89 \pm 0.07$ |
| | DD (↓) | $9.41\,(0;23)$ | $21.7\,(0;71)$ | $17.8\,(11;27)$ | $-\,(-;-)$ | $-\,(-;-)$ | $7.5\,(3;23)$ | $\mathbf{6.5}\,(2;16)$ | $\mathbf{6.2}\,(0;26)$ | $11.8\,(8;14.5)$ |
| | FP (↓) | $0.4\,(0;1)$ | $0.66\,(0;2)$ | $0.93\,(0;2)$ | $-\,(-;-)$ | $-\,(-;-)$ | $0.4\,(0;2)$ | $0.33\,(0;1)$ | $0.4\,(0;1)$ | $\mathbf{0.13}\,(0;1)$ |
| HASC | AUC (↑) | $0.73 \pm 0.12$ | $0.66 \pm 0.14$ | $0.65 \pm 0.10$ | $0.84 \pm 0.15$ | $0.75 \pm 0.16$ | $0.81 \pm 0.13$ | $\underline{0.85} \pm 0.12$ | $0.79 \pm 0.2$ | $\mathbf{0.87} \pm 0.12$ |
| | COV (↑) | $0.57 \pm 0.19$ | $0.59 \pm 0.32$ | $0.66 \pm 0.24$ | $\mathbf{0.79} \pm 0.18$ | $0.66 \pm 0.26$ | $0.75 \pm 0.10$ | $\underline{0.78} \pm 0.13$ | $0.75 \pm 0.25$ | $\underline{0.78} \pm 0.19$ |
| | DD (↓) | $357\,(0;1264)$ | $334\,(0;1540)$ | $445\,(0;1866)$ | $180\,(0;1054)$ | $559\,(3.5;4040)$ | $496\,(0;3678)$ | $454\,(0;4006)$ | $233\,(0;1342)$ | $\mathbf{39}\,(0;688)$ |
| | FP (↓) | $3.8\,(0;8)$ | $14\,(0;47)$ | $9.0\,(0;46)$ | $0.78\,(0;4)$ | $4.7\,(0;24)$ | $1.5\,(0;5)$ | $1.3\,(0;4)$ | $3.7\,(0;18)$ | $\mathbf{0.09}\,(0;1)$ |
| HAR | AUC (↑) | $0.82 \pm 0.07$ | $\mathbf{0.85} \pm 0.06$ | $0.76 \pm 0.06$ | $0.53 \pm 0.05$ | $0.72 \pm 0.09$ | $0.81 \pm 0.1$ | $0.80 \pm 0.06$ | $0.73 \pm 0.06$ | $\mathbf{0.85} \pm 0.07$ |
| | COV (↑) | $0.76 \pm 0.12$ | $\mathbf{0.82} \pm 0.07$ | $0.53 \pm 0.09$ | $0.11 \pm 0.04$ | $0.54 \pm 0.06$ | $0.67 \pm 0.08$ | $0.66 \pm 0.07$ | $0.52 \pm 0.07$ | $0.56 \pm 0.04$ |
| | DD (↓) | $4.7\,(1.25;9.3)$ | $3.7\,(1.0;7.7)$ | $2.8\,(1.8;4.2)$ | $10.3\,(9;12)$ | $7.1\,(4.9;9.1)$ | $3.2\,(1.1;6.5)$ | $3.4\,(1;5.9)$ | $\mathbf{1.8}\,(0.5;4.2)$ | $4.8\,(2.8;6.5)$ |
| | FP (↓) | $4.9\,(1;14)$ | $2.5\,(0;8)$ | $\mathbf{0.1}\,(0;1)$ | $0.33\,(0;1)$ | $2.2\,(0;4)$ | $0.7\,(0;3)$ | $0.8\,(0;2)$ | $0.2\,(0;1)$ | $\mathbf{0.1}\,(0;1)$ |

**Occupancy:** This dataset is designed for the task of detecting changes in office occupancy levels based on various room condition measurements, and is commonly used for the evaluation of change point detection methods Van den Burg & Williams (2020). Originally, it was introduced in Candanedo & Feldheim (2016) and captures four different measurements: 1) temperature, 2) humidity level, 3) light, and 4) $CO_2$. While SWCPD and ClaSP show the best results for the AUC scores, SWCPD additionally delivers strong Covering scores, and minimal false Positives. For the results in Table 4, we set $\tau = 30$, $w = 500$, $K_{\max} = 500$, $L = 1000$, $p = 2$, and $q = 0.05$.

---

[1]Best performance is determined after applying a paired t-test, bold numbers indicate best absolute performance, underlined numbers indicate equal performance with a smaller reported metric.

## 5 LIMITATIONS

Despite the demonstrated effectiveness of SWCPD, several limitations merit attention. First, the reliance on random one-dimensional projections can reduce sensitivity to subtle, local changes in high-dimensional spaces, as these may not always be captured by a limited sampling of directions. Future refinements might involve adaptive or learned projection strategies that more selectively probe feature dimensions most likely to exhibit drift. Second, our adaptive thresholding scheme is based on the theoretically derived Gamma-distribution of Sliced Wasserstein distances; in practice, however, for smaller datasets or heavy-tailed data can undermine our theoretical approximation.

## 6 CONCLUSION

We introduced SWCPD, a novel framework for explainable online change point detection in high-dimensional data streams, leveraging Sliced Wasserstein (SW) distance. By transforming multivariate time series into a one-dimensional signal, our method circumvents the computational bottlenecks of traditional CPD techniques. We integrated three key innovations: (1) a statistically grounded SW-based transformation that enables CPD on high-dimensional data with minimal overhead, (2) a self-adaptive thresholding mechanism that dynamically calibrates detection sensitivity using a Gamma-based statistical hypothesis test, and (3) a contrastive explainability module that identifies the most influential feature dimensions contributing to detected changes.

We demonstrated SWCPD's superiority across multiple benchmarks, achieving competitive detection performance while maintaining interpretability. SWCPD outperforms existing online and offline CPD techniques, particularly in dynamic, high-dimensional settings where both reliability and explainability are critical. The proposed feature attribution mechanism offers actionable insights by revealing the root causes of distributional shifts, ensuring that detected changes are not only statistically significant but also interpretable.

SWCPD is a practical bridge between modern data streams and the social-technical systems that rely on them. Interpretable, distribution-level telemetry is quickly becoming as mission-critical as traditional point-estimate monitoring. As interpretable, distribution-level telemetry becomes as critical as point-estimate monitoring, SWCPD combines statistical rigor with human-centered explanations. This enables downstream AI systems, from LLMs to sensor stacks, to be wrapped in transparent "change firewalls," promoting a future where real-time models both detect and justify shifts, setting a new standard for safer, fairer, and more accountable AI.

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

Table 5: Parameter setting ViT

| BATCH SIZE | EPOCHS | LR | PATCHSIZE | DIM | DEPTH | HEADS | MLP |
|---|---|---|---|---|---|---|---|
| 64 | 15 | $1 \times 10^{-4}$ | 4 | 64 | 6 | 8 | 128 |

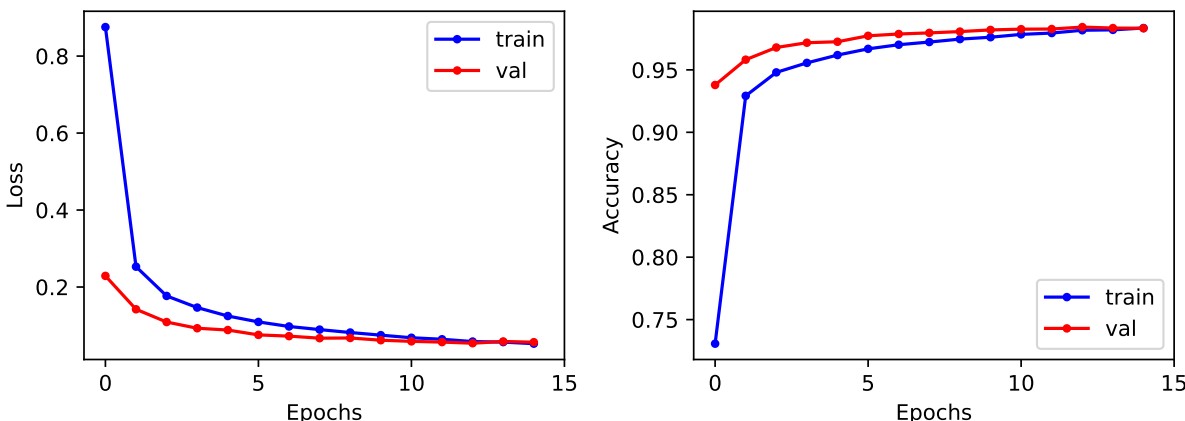

Figure 4: Illustrates Train and validation curves of loss and accuracy over 15 epochs for ViT model.

# A  APPENDIX

# B  ADDITIONAL EXPERIMENTS

All experiments were conducted on a machine equipped with an AMD Ryzen 7 5700X CPU, 32 GB of RAM, and a RTX 3060 GPU.

## B.1  EXPLAINABILITY

### B.1.1  MNIST

**Vision Transformer.** We employ a Vision Transformer (ViT) model for image classification on the MNIST dataset. The model processes input images of size $28 \times 28$ pixels, which are divided into non-overlapping patches of size $4 \times 4$, resulting in 49 patches. Each patch is linearly embedded into a 64-dimensional feature space. The transformer consists of 6 layers, each employing multi-head self-attention with 8 heads and a feed-forward network with a hidden dimension of 128. We apply a dropout rate of 0.1 during the embedding and transformer layers to prevent overfitting. Since MNIST images are grayscale, the model is configured to accept single-channel input. The data was split into $90\%$ training set of which $10\%$ into the validation set, while we used the additional $10\%$ for testing. We use Adam with $\lambda = 0.001$ for training over 15 epochs with a batch size of 64.

**CNN.** We use a simple LeNet-5 LeCun et al. (1998) as a benchmark CNN to investigate model explanations under drifts on MNIST. We use the same train-test split as for the ViT model and Adam optimizer with step size $\lambda = 0.001$. We repeat the same procedure as for the ViT and introduce drifts and investigate the differences in the feature attrituions using SWD, and SoTA explanations methods IG, GS, and DL. From fig. 5, we see that all reference methods align with feature attributions, and hence show the same pattern for differences of before and after drift. Although, all explanation methods align with the most significant feature changes, the pixelwise distance based approach (SWD) narrows them down the most. This can also be seen in fig. 6, which highlights the differences of adversarial examples changing the model output between two given classes, as SWD shows a strong alignment.

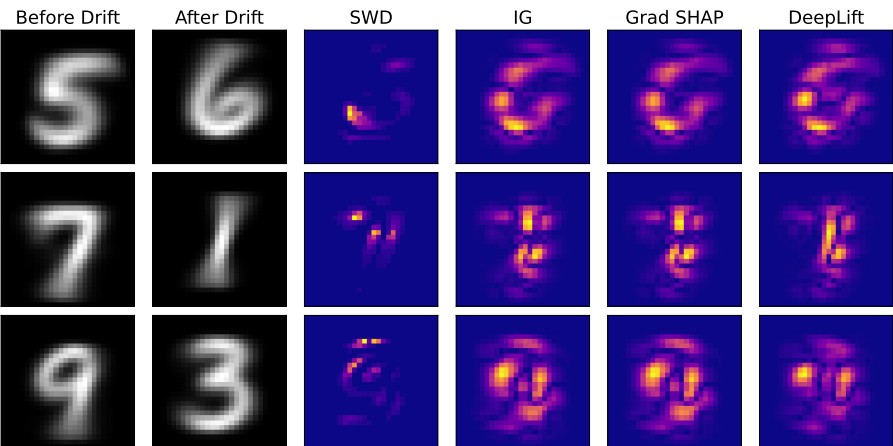

Figure 5: Shows the absolute difference of mean feature attributions for three different drifts and reference methods IG, GS, and DL.

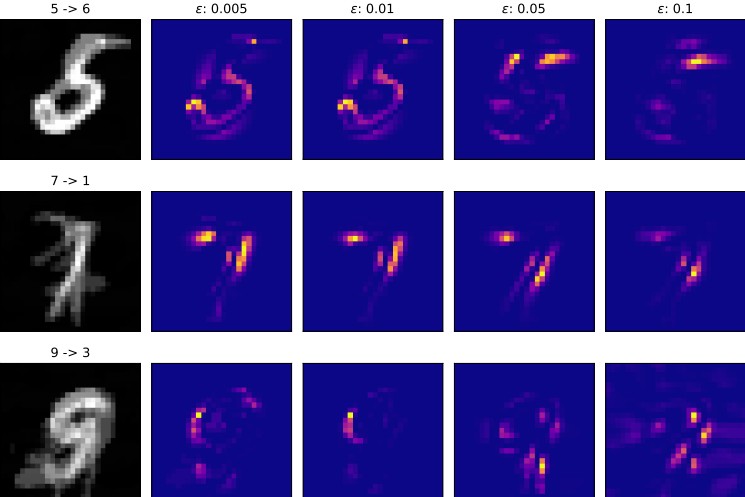

Figure 6: Shows mean adversarial examples (left) which changes the model (CNN) output from $5 \rightarrow 6$, $7 \rightarrow 1$, and $9 \rightarrow 3$ using FGSM for different $\epsilon$, and $L_4$-norm between mean adversarial example and non-adversarial example

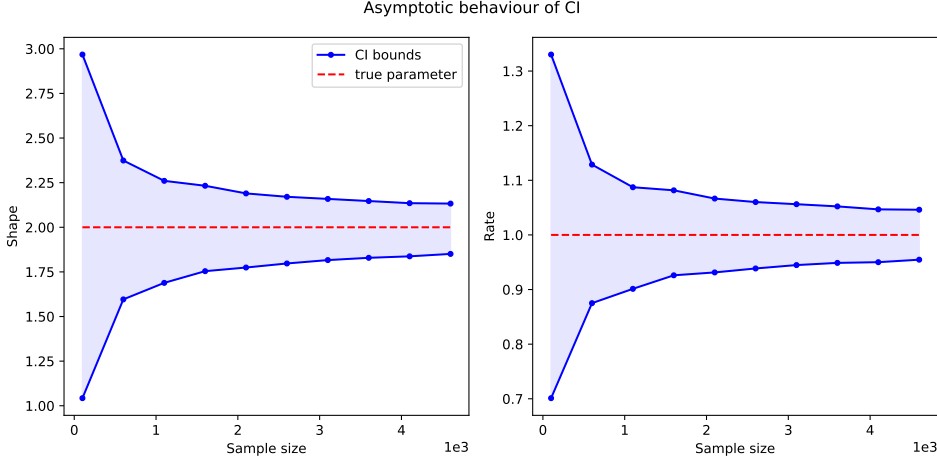

Figure 7: Shows the lower and upper bound of confidence interval (eq. (3)) for MoM estimator $\hat{\alpha}, \hat{\beta}$ averaged over 30 experiments for equidistant sample sizes from $n = 100, \ldots, 5000$.

### B.2 UNCERTAINTY QUANTIFICATION

We investigate the asymptotic behaviour of the confidence intervals obtained by theorem 3.2 for $X \sim \Gamma(2, 1)$ for various sample sizes and calculate the average confidence intervals for 30 different random samples $X_n$ with sample size $n$. For an increasing sample size, the confidence intervals for both parameters shrinks and is centered around the true parameters as expected since sample mean and variance are consistent, see fig. 7.

### B.3 DISTRIBUTION OF RANDOM PROJECTIONS

For the numerical study of the distribution of $w_2^2(\theta) : \theta \mapsto \mathrm{W}_2(\mathbb{P}^\theta, \mathbb{Q}^\theta)$, we consider two sample sets $X, Y$ each consisting of 200 MNIST samples with gray-scaled images from the same class respectively. For this example we set the class of each sample from $X$ to 1, and $Y$ to 7. We calculated the SWD between both samples for different numbers of random projections ranging from $L = 100, 500, 1000, 5000$. We then constructed the MoM esitmates of a Gamma distribution based on the set of random projection obtained. Furthermore, we calculated a Kernel density estimation for the random projections itself. This shows that using a Gamma distribution indeed fits the data obtained. Additionally, we compared the sampled quantiles and the theoretical quantiles of the random projections and MoM fitted Gamma distribution to asses the goodness of fit. The result is summarize in fig. 8, as expected, we see that as the number of projection increases, we obtain a better fit. While fig. 8, shows the asymptotic behaviour given by Theorem 3.1 of the linear random projections of the Sliced Wasserstein distance, we observed that it also holds for lower-dimensional data, e.g. simulated synthetic data. Consider $x \in \mathbb{R}^d$, we fix a projection direction $\theta_l \sim \mathcal{U}(S^{d-1})$ and consider a sample set $X = (x_1, x_2, \ldots, x_n)$. We set $z_l = \langle X, \theta_l \rangle$, where $z_l$ is normal due to the CLT for $d \to \infty$. We simulated $x$ according to $d$ independent exponential distributions $\lambda = 1$ and applied the Sharpio-Wilk test Shapiro & Wilk (1965) to asses wheter the projected samples can be considered normal distributed. In table 6, we report the average $p$-values projections obtained using $L \in [100, 500, 1000]$ for various dimensions $d$.

**Approximation Error:** We now report the Mean Absolute Error (MAE) between the theoretical quantiles and observed quantiles. The theoretical quantiles are derived from a Gamma distribution based on the MoM estimates from the random projections involved in the calculation of the Sliced Wasserstein distance. The observed quantiles are calculated based on the empirical distribution of the random projections. For each dimension, we simulate two independent datastreams, each consisting of $d$ independent Gaussian distributions with a uniformly sampled mean. We vary $d$ and $L$, use fixed random seeds, and report the results for 10 trials in Table 7.

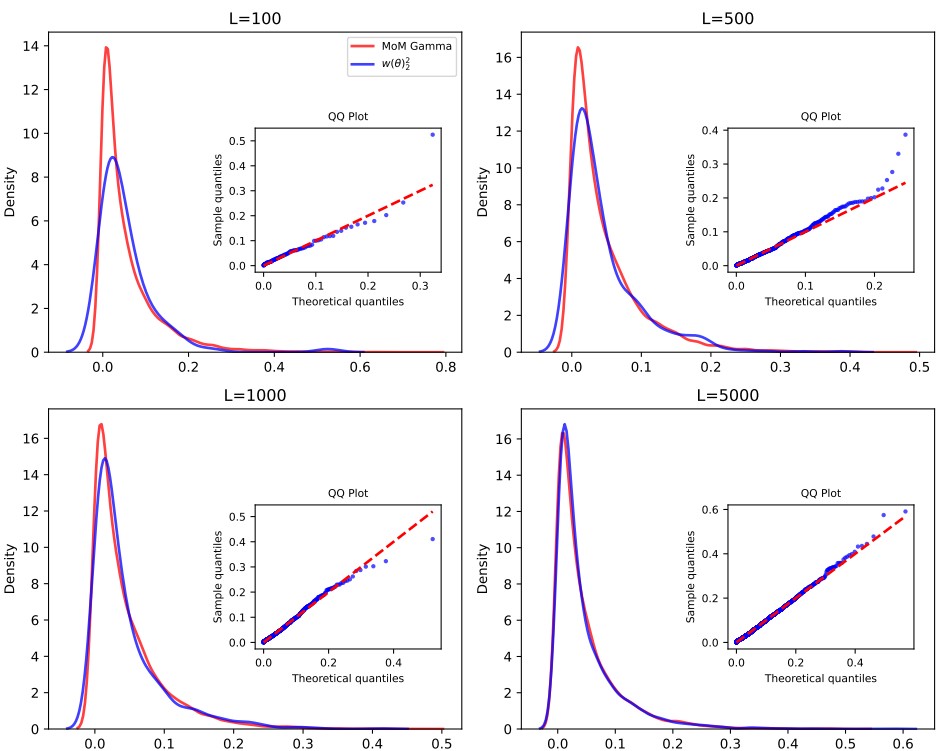

Figure 8: Shows a Kernel density estimation of a gamma density using the MoM estimated parameters (red line) for the random projection for various number of projections $L = 100, 500, 1000, 5000$, and the KDE of random projections (blue line) itself between two samples from MNIST.

Table 6: Average $p$-values obtained using Sharpio-Wilk test

|  | $L$ | | |
| --- | --- | --- | --- |
| $d$ | 100 | 500 | 1000 |
| 10 | 0.44 (✓) | 0.065 (✓) | 0.005 (-) |
| 20 | 0.5 (✓) | 0.3 (✓) | 0.2 (✓) |
| 30 | 0.5 (✓) | 0.4 (✓) | 0.3 (✓) |
| 60 | 0.5 (✓) | 0.5 (✓) | 0.5 (✓) |
| 100 | 0.5 (✓) | 0.5 (✓) | 0.5 (✓) |

Table 7: Shows MAE between theoretical and observed quantiles of a Gamma distribution derived from 2-Wasserstein distance between random projections.

| $d$ | $L = 100$ | $L = 1.000$ | $L = 10.000$ |
| --- | --- | --- | --- |
| 5 | $1.12 \pm 0.17$ | $0.35 \pm 0.03$ | $0.37 \pm 0.01$ |
| 10 | $0.345 \pm 0.06$ | $0.31 \pm 0.06$ | $0.15 \pm 0.01$ |
| 20 | $0.401 \pm 0.07$ | $0.16 \pm 0.03$ | $0.02 \pm 0.01$ |
| 100 | $0.291 \pm 0.05$ | $0.11 \pm 0.01$ | $0.04 \pm 0.01$ |
| 200 | $0.298 \pm 0.05$ | $0.13 \pm 0.04$ | $0.04 \pm 0.01$ |

### B.4 STOPPING CRITERION IN ALGORITHM 2

In Algorithm 2, we update the removed feature from $Y$ with samples $X$. Suppose, we have observations $X_1, \ldots, X_N \sim P_X$, and $Y_1, \ldots, Y_N \sim P_Y$. Without any drifted components, we have $P_X = P_Y$ with

$$m = \mathbb{E}[X] = \mathbb{E}[Y] \in \mathbb{R}$$

$$\Sigma = \text{Cov}(X) = \text{Cov}(Y) \in S_+^d$$

where $m_X = \frac{1}{N} \sum_i^N X_i$, and $m_Y = \frac{1}{N} \sum_{i=1}^N Y_i$ denote the sample means and $S_+^d$ denotes the set of symmetric p.s.d. $d \times d$ matrices. We consider

$$||D|| = ||m(X) - m(Y)||,$$

then

$$\mathbb{E}[||D||] \leq \sqrt{\frac{2}{N} tr(\Sigma)}$$

Since we have $D \sim \mathcal{N}(0, \frac{2}{N}\Sigma)$, we can decompose $\Sigma = U\Lambda U^T$. Then with $Z = U^T D$, it follows $Z \sim \mathcal{N}(0, \frac{2}{N}\Lambda)$. Thus $||D||^2 = \sum_{i=1}^d \frac{2}{N}\lambda_i \chi_1^2$, with $\chi_1^2$ denotes a chi-squared distribution with one degree of freedom. Note that $tr(\Sigma) = \sum_{i=1}^d \lambda_i$, where $\lambda_i$ is the $i$-th eigenvalue for $i = 1, \ldots, d$. Therefore, we have

$$\mathbb{E}||D||^2 = \frac{2}{N} tr(\Sigma),$$

applying Jensen inequality yields

$$\mathbb{E}[||D||] \leq \sqrt{\frac{2}{N} tr(\Sigma)}.$$

## C CHANGE DETECTION

### C.1 ABLATION STUDY

In the following we are going to investigate the sensitivity and influence of SWCPD for variations in its key hyperparameters. Our proposed method relies on the following hyperparameter:

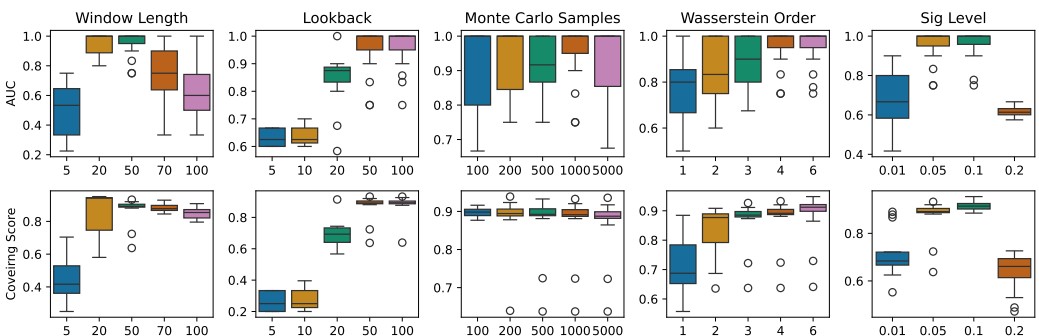

Figure 9: Shows boxplots of the AUC and Covering scores for each parameter variations while keeping the other parameters fixed.

- $L = 500$: Number of random projections (Monte Carlo samples)

- $w = 50$: Window length

- $p = 2$: Order of Wasserstein distance

- $q = 0.05$ : Significance level

- $K_{max} = k$: Maximum length of lookback window (for moving average calculation)

We conducted experiments using the same MNIST datasets as in the experimental section of the paper, hence the number of change points varies from 2 to 4 with 200 samples for each sub-sequence forming one segment. We defined the following parameter sets, $w \in [5, 20, 50, 70, 100]$, $K_{max} \in [5, 10, 20, 50, 100]$, $L \in [100, 200, 500, 1000, 5000]$, $p \in [1, 2, 3, 4, 6]$, and $q \in [0.01, 0.05, 0.1, 0.2]$. Across all simulation on all 15 datasets, we fixed the random seed for the Monte Carlo samples to obtain reproducible results. We choose the default parameter $L = 5000$, $p = 4$, $w = 50$, $K_{max} = 50$, $q = 0.05$ which we fixed, only varying one parameter within its parameter set respectively. Figure 9 shows the parameter sensitivity of SWCPD for this exemplary dataset. This shows, that the most sensitive parameter are the window length, and lookback window, whereas the number of Monte Carlo samples may be sufficiently large if chosen $L \approx d$. The Wasserstein order should be set above 2, depending on the severity of the drifts, since it amplifies low signals (small distances). The same holds for the significance level as it may be irrelevant if the abrupt changes are significant itself. To further emphasize the influence of the Wasserstein order and significance level, we run additional experiments on synthetic datasets with low drift severities. We used the sampling scheme described in section C.2.1, where we set $N = 1500$, $d = 10$ with initial base center $c_0 \in [-4, 4]^{10}$ and 10 different segments. We selected $\mathcal{V} = \{1, 2, 3\}$ and drift severity was set to $\delta_j \sim \text{Uniform}(-1)$ for each feature index in $\mathcal{V}$. In contrast we sampled the remaining data with i.i.d. Gaussian distribution with mean at each base center respectively and $\sigma = 0.5$ for each component. The result highlights the influence of the significance level for the propagated upper bound as increasing the variable leads to a decrease in the AUC and Covering score since the number of false negatives increases when the upper bound is to close to the cumulative sum. In this example, the Wasserstein order was of secondary importance as changing it lead to similar scores across the datasets, however increasing the Wasserstein order has a smoothing effect on the cumulative sum as small Wasserstein distances nearly vanishes. This can be benefiting for noisy signals. For weak signals, where the abrupt changes are small, we suggest decreasing the Wasserstein order amplifying small changes in the underlying data. Additionally, we performed a Grid Search on MNIST and Occupancy. For both experiments, we fixed $p = 4$, $L = 5000$ while varying the significance level $q$, window size $w$, and Lookback $K_{max}$. We limited the possible parameter values for MNIST to $w \in [20, 30, 40, 50, 100]$, $K_{max} = [0.5w, w]$, and $q = [0.01, 0.05, 0.1]$. We report the average AUC scores for each parameter combination in fig. 11, we see multiple parameter sets achieving high AUC scores. For Occupancy, we limited the possible parameter values to $w \in [200, 300, 400, 500, 600]$, $K_{max} = [0.25w, 0.5w, 0.75w, w]$, and $q = [0.01, 0.05, 0.1]$. We report the AUC scores for each parameter combination in fig. 12, we see multiple parameter sets achieving high AUC scores in comparison to the baseline methods.

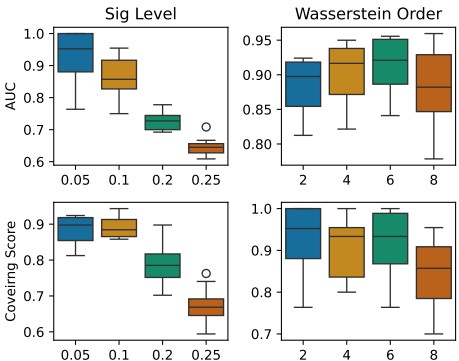

Figure 10: Summary of AUC and Covering scores for varying significance level and Wasserstein order on 10 different synthetic datasets with $d = 10$, $N = 1500$ and 10 drifts in 3 features simultaneously.

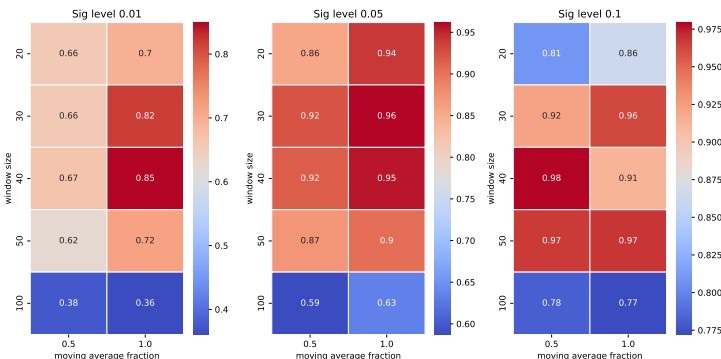

Figure 11: Average AUC scores for various parameter combinations using SWCPD on MNIST sequences.

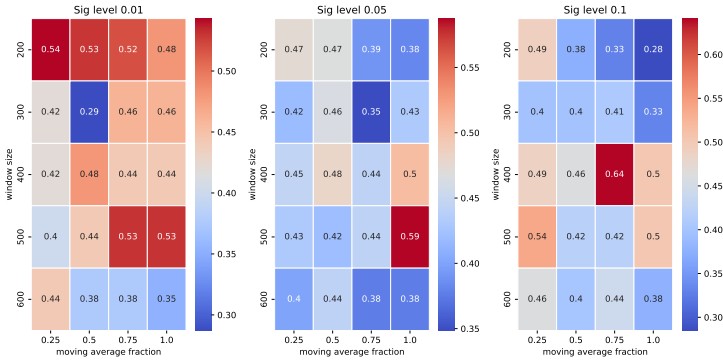

Figure 12: AUC scores for various parameter combinations using SWCPD on Occupancy.

## C.2 METHODS

In the following part, we will describe the reference methods used within the Change Point Detection experiments. Alongside its main parameters and their default values, we also describe the setting for each dataset. We provide an overview of the computational complexity in Table 8.

Table 8: Overview of reference methods and respective time complexity for online and offline change point detection, $K$: number of change points, $d$: dimension, $N$: total samples, $w$: sliding window.

| Method | parametric | non parametric | online | offline | Offline Complexity[2] | Online Complexity[3] |
|---|---|---|---|---|---|---|
| e-divisive | (✓) | | | (✓) | $\mathcal{O}(KN^2)$ | $\mathcal{O}(KN^4)$ |
| KCP | | (✓) | | (✓) | $\mathcal{O}(KdN^2)$ | $\mathcal{O}(KdN^4)$ |
| ClaSP | | (✓) | | (✓) | $\mathcal{O}(KN^2)$ | $\mathcal{O}(KN^4)$ |
| BOCPD | (✓) | | (✓) | | $(-)$ | $\mathcal{O}(Nd)$ |
| OT-CPD | | (✓) | | (✓) | $\mathcal{O}(N(w^3\log(w) + w^2d))$ | $\mathcal{O}(N(w^3\log(w) + w^2d))$ |
| SWCPD (ours) | | (✓) | (✓) | | $(-)$ | $\mathcal{O}(N(wdL + Lw\log w))$ |

**BOCPD (online):**  Bayesian Online Change Point Detection (BOCPD) Adams & MacKay (2007) is a method used to detect change points in streaming data in real time. It has some desirable properties, such that it can be applied online, is applicable to multivariate data, and quantifies uncertainty Knoblauch & Damoulas (2018). The underlying concept of this approach is to monitor the probability of a change point occurring at each time step by maintaining and updating the posterior distribution over potential segmentations of the data. It assumes that data within a segment follows a consistent probabilistic model (e.g., Gaussian), and a change point indicates a shift in the underlying model. There exist many implementation, we use the implementation that comes with the `ocp` package Pagotto (2019). The key parameters for this method are:

- `prob_model`: the underlying probability model of the posterior distribution

- `init_params`: the initial parameters for the probability model consiting of $m, k, a, b$

- `hazard_function`: normally set to a constant function with certain hazard rate $\lambda$

We run the experiments with the following parameter sets:

- **HASC**
    - `prob_model` : "$gaussian$"
    - `init_params` : $m = 0, k = 10, a = 0.1, b = 0.01$
    - `hazard_function` : type=constant, $\lambda = 100$
- **HAR**
    - `prob_model` : "$gaussian$"
    - `init_params` : $m = 0, k = 0.01, a = 0.01, b = 1e - 4$
    - `hazard_function` : type=constant, $\lambda = 100$
- **MNIST**
    - `prob_model` : "$gaussian$"
    - `init_params` : $m = 0.3, k = 0.01, a = 0.01, b = 1e - 4$
    - `hazard_function` : type=constant, $\lambda = 100$
- **Occupancy**
    - We additionally applied z-score normalization of the data beforehand to obtain a reasonable distributional setting and obtain change points
    - `prob_model` : "$gaussian$"
    - `init_params` : $m = 0, k = 0.01, a = 0.01, b = 1e - 4$
    - `hazard_function` : type=constant, $\lambda = 100$

---

[2]Complexity for offline change point detection for a multivariate time series with $d$ dimensions and $N$ observations

[3]Accrued complexity for change point detection at time step $t = N$ for a multivariate time series with $d$ dimensions and in total $N$ observations

**E-divisive (offline):** The e-divisive combines binary bisection together with a permutation test based on an energy divergence measure Matteson & James (2014). It is a non-parametric offline change point detection method for multivariate data, making it applicable to a wide range of complex data. We use the implementation from the `ecp` package Nicholas A. James et al. (2019). The method relies on the following parameters with default specification:

- `R` $= 199$ : specifies the number of permutations test applied
- `sig.lvl` $= 0.05$ : the significance level of the permutation test
- `min.size` $= 30$ : the minimum observations between two subsequent change points

We run the experiments with the following parameter sets:

- **HASC:** `R` $= 199$, `sig.lvl` $= 0.05$, `min.size` $= 500$
- **HAR:** `R` $= 199$, `sig.lvl` $= 0.05$, `min.size` $= 30$
- **MNIST:** `R` $= 199$, `sig.lvl` $= 0.05$, `min.size` $= 30$
- **Occupancy:** `R` $= 30$, `sig.lvl` $= 0.05$, `min.size` $= 400$

**KCP (offline):** Kernel change-point detection (KCP) transforms the data into a RKHS with an associated kernel, which is used to calculate the dissimilarity (cost). The goal is to obtain an optimal segmentation of the input data in the sense of a minimized averaged cost within each segment obtained Arlot et al. (2019). An efficient implementation of this method can be found in Truong et al. (2020), we assume that the number of change points is unknown, hence we rely on `KerneCPD` with `PELT`. The methods relies on the following parameter:

- `kernel` $= "linear"$: specifies the kernel, cost function
- `min_size` $= 1$: minimum segmentation length
- `pen`: penalty or regularization of number of change points identified

The penalty value needs to be specified if the number of change point is unknown. Usually a higher value will lead to fewer change points identified, while a lower value encourages the method to annotate more change point with a more fine grained segmentation. We used the following parameter settings:

- **HASC:** `kernel` $=$"rbf", `min_size` $= 2$, `pen` $= 10$
- **HAR:** `kernel` $=$"rbf", `min_size` $= 2$, `pen` $= 1$
- **MNIST:** `kernel` $=$"rbf", `min_size` $= 2$, `pen` $= 1$
- **Occupancy:** `kernel` $=$"rbf", `min_size` $= 2$, `pen` $= 50$

**ClaSP (offline):** ClaSP (Classification Score Profile) is a self-supervised time series segmentation method Ermshaus et al. (2023b). The implementation is available at `https://github.com/ermshaua/claspy`. It is a dynamic windowing approach which creates a binary classification problem across different split points of the time series using $k$-Nearest Neighbors (k-NN) which is evaluated using corss validation. The score obtained from k-NN is used to evaluate the similarity of both segments, where higher scores indicate a stronger dissimilarity. The main parameters to choose are:

- `windwo_size` $= "suss"$: size of the sliding window, default Summary Statistics Subsequence (suss)
- `k_neighours` $= 3$: number of nearest neighbours for k-NN
- `distance` $= "znormed\_euclidean\_distance"$: distance used for k-NN

We used the following parameters:

- **HASC:** `windwo_size` $= 50$
- **HAR:** `windwo_size` $= 30$
- **MNIST:** `windwo_size` $= 100$
- **Occupancy:** `windwo_size` $= 30$

**OT-CPD (offline):**   OT-CPD Cheng et al. (2020a) is a optimal transport based change point detection method which calculates the Wasserstein distance between two sliding windows. After obtaining all available data, it applies a matched filter on the Wasserstein test statistic to obtain a more persistent test statistic reducing false positives. OT-CPD annotates a change if the filtered test statistic exceeds a pre-defined threshold. In our experiments, we relied on the implementation available at `https://github.com/kevin-c-cheng/OtChangePointDetection/tree/master`. The main parameters for the change point detection method to choose are:

- `window`: size of the sliding window

We used the following parameters:

- **HASC:** $\text{window} = 1000$
- **HAR:** $\text{window} = 25$
- **MNIST:** $\text{window} = 150$
- **Occupancy:** $\text{window} = 750$

**RuLIFS:**   Relative unconstrained least-squares importance fitting (RuLSIF) estimates a relative density ratio that mixes the two distributions using a parameter $\alpha$. The relative ratio is approximated using a kernel model, and its parameters are obtained by solving a simple least-squares problem with a closed form solution. From this estimated ratio, the method computes a divergence score that becomes large when the two windows differ. In a sliding window approach this scores is computed for which peaks indicate change points. The main parameters for the change point detection method to choose are:

- $\alpha$: mixture coefficient in $\alpha$-relative density ratio
- `window`: size of the sliding window
- `kernel_num`: number of kernels used
- `steps`: stride of sliding window

We used the following parameters:

- **HASC:** $\text{window} = 200, \alpha = 0.1, \text{kernel\_num} = 10$
- **HAR:** $\text{window} = 20, \alpha = 0.1, \text{kernel\_num} = 10$
- **MNIST:** $\text{window} = 100, \alpha = 0.1, \text{kernel\_num} = 10$
- **Occupancy:** $\text{window} = 250, \alpha = 0.1, \text{kernel\_num} = 10$

**DeepRuLIFS:**   DeepRuLIFS Hushchyn et al. (2020); Hushchyn & Ustyuzhanin (2021) follows the framework of RuLIFS where the $\alpha$ relative density ratio is estimated using a deep neuronal network. We rely on the implementation given by [4]. The main parameter for the change point detection method which we varied where:

- `lag_size` : the gap between batches

All other hyperparameter were kept as default. We used the following parameters:

- **HASC:** $\text{lag} = 250$
- **HAR:** $\text{lag} = 20$
- **MNIST:** $\text{lag} = 100$
- **Occupancy:** $\text{lag} = 250$

---

[4]`https://gitlab.com/lambda-hse/change-point/online-nn-cpd`

**DeepCLF:** This method trains a neuronal network to distinguish a reference window from a more test window based on a divergence metric. By sliding the windows forward in time and measuring their divergence, peaks in the score curve reveal where the underlying data distribution has changed Hushchyn et al. (2020). The main parameter for the change point detection method which we varied where:

- `lag_size` : the gap between batches

All other hyperparameter were kept as default. We used the following parameters:

- **HASC:** `lag` = 250

- **HAR:** `lag` = 20

- **MNIST:** `lag` = 100

- **Occupancy:** `lag` = 250

### C.2.1 SYNTHETIC DATA

The proposed sampling scheme generates synthetic data with customizable cluster centers and variable feature dimensions. The process begins by defining an initial base center $\mathbf{c}_0 \in \mathbb{R}^d$, where $d$ is the number of features. This base center serves as the reference point for all subsequent cluster centers.

To generate additional cluster centers, a perturbation process is applied to $\mathbf{c}_0$. Specifically, for each new cluster center $\mathbf{c}_i$, $i = 1, \ldots, k-1$, the following transformation is applied:

$$c_{i,j} = \begin{cases} c_{0,j} + \Delta_j & \text{if } j \in \mathcal{V}, \\ c_{0,j} & \text{otherwise}, \end{cases}$$

where $c_{i,j}$ is the $j$-th feature of the $i$-th cluster center, $\mathcal{V} \subseteq \{1, 2, \ldots, d\}$ is the set of varying feature indices, and $\Delta_j \sim \text{Uniform}(-\delta, \delta)$ is a random offset sampled from a uniform distribution with range $[-\delta, \delta]$.

The sampling process ensures that only the features indexed by $\mathcal{V}$ are modified, while other features remain constant across all cluster centers. After generating the cluster centers, the data points are sampled from a multivariate Gaussian distribution. For each cluster $i$, the samples $\mathbf{x}_i^{(n)}$, $n = 1, \ldots, N_i$, are drawn as:

$$\mathbf{x}_i^{(n)} \sim \mathcal{N}(\mathbf{c}_i, \Sigma),$$

where $\Sigma \in \mathbb{R}^{d \times d}$ is the covariance matrix (diagonal for simplicity) and $N_i$ is the number of samples assigned to cluster $i$. The total number of samples $N$ is distributed evenly across clusters, i.e., $N_i = N/k$.

This scheme allows for precise control over the features that vary between groups $\mathcal{V}$, the degree of variation $\delta$, and the variance of data points within each cluster with $\Sigma$. By adjusting these parameters, synthetic datasets can be tailored for specific experimental purposes, such as evaluating clustering algorithms or analyzing feature-specific effects. In Table 9 we report AUC scores for different variances and drift severities for Gaussian synthetic data with $d = 10$ and 1500 samples with 3 segments. Additionally, Figure 13 illustrates the contrastive explanations for the obtained change points by SWCPD. We set the window length $w = 50$, the lookback window for the estimation of shape- and rate parameters $K_{\max} = 50$, $p = 2$, and $L = 5000$.

Table 9: AUC for different variances $\sigma^2$ and drift severity $|\delta|$

| Source | Value | $\tau = 5$ | $\tau = 10$ | $\tau = 20$ |
|---|---|---|---|---|
| | 0.1 | $1.0 \pm 0.0$ | $1.0 \pm 0.0$ | $1.0 \pm 0.0$ |
| Variance ($\sigma^2$) | 0.5 | $0.8 \pm 0.28$ | $0.93 \pm 0.14$ | $1.0 \pm 0.0$ |
| | 1.0 | $0.65 \pm 0.32$ | $0.75 \pm 0.29$ | $0.91 \pm 0.13$ |
| | 1 | $0.4 \pm 0.15$ | $0.6 \pm 0.26$ | $0.94 \pm 0.08$ |
| Drift Severity ($|\delta|$) | 2 | $0.6 \pm 0.22$ | $0.8 \pm 0.27$ | $0.97 \pm 0.06$ |
| | 3 | $0.71 \pm 0.28$ | $0.87 \pm 0.24$ | $0.98 \pm 0.05$ |

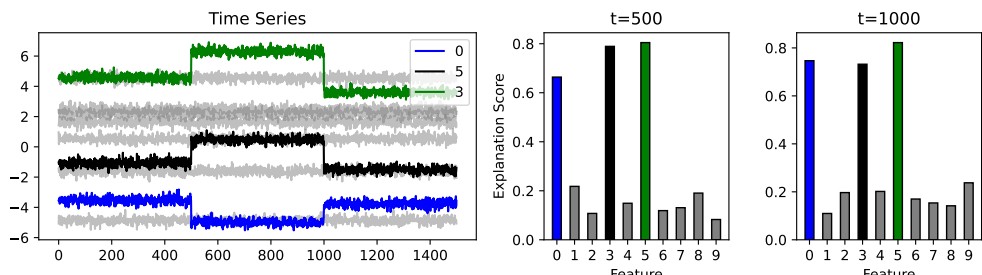

Figure 13: Interpretable change points obtained with SWCDP. Two right plots show feature attributions obtained using Algorithm 2, showing alignment with ground truth root causes of the drifts.

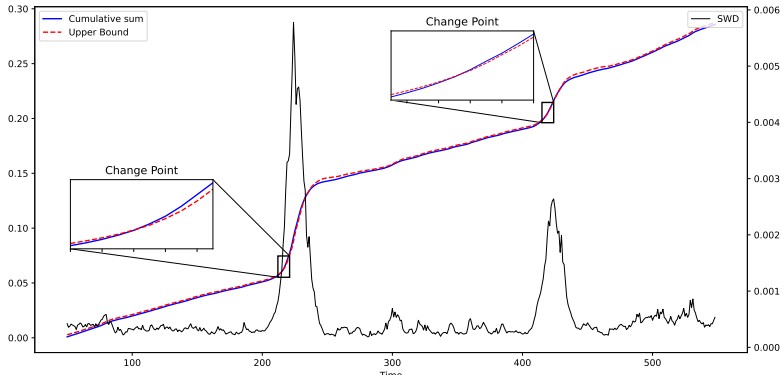

Figure 14: Visualizes our proposed detection method for MNIST data with two change points at $t = 200, 400$. Change points are indicated when the cumulative sum exceed the upper bound which is derived based on past SWDs.

### C.2.2 MNIST

In order mimic a streaming behaviour, we uniformly sample an initial class (without replacement) and select $K$ instances from the current class. We repeat this procedure and annotate the samples to introduce abrupt changes. Within the scope of the experiments for this paper, we generated 5 distinct data sequences with $2, 3,$ and $4$ change points, where each class has 200 samples. We illustrate SWCPDs detection procedure for a sampled MNIST sequence with two change points at $t = 200, 400$ in fig. 14. By calculating tracking the SW distance using a rolling window of $k = 50$ observations, we obtain a one-dimensional signal with two significant spikes at $t_1 = 225$ and $t_2 = 425$ since the within similarity of the rolling window will be the largest when the first half samples belong to class prior to the drift and the second half to the class after the drift. We see, that using a propagated upper bound given the current state instead of purely relying on the distance as a signal, we can anticipate changes more reliable and faster. Moreover, the upper bound is adaptive such that there is no fine tuning or manually shifting the rolling window involved. SWCPD is based on the Sliced Wasserstein distance which is a metric from Optimal Transport (OT). To contextualize the computational performance of our proposed method for other OT-based detection methods such as OT-CPD, and e-divisive, we report the average wall-clock time and standard deviation in Table 10.

### C.2.3 HASC

The dataset consists of distinct multimodal multivariate time series monitoring human motion of different daily activities. The data was collected as part of the Human Activity Segmentation Challenge Ermshaus et al. (2023a) using built-in smartphone sensors. In total, the dataset has 250 time series consisting of 12 different measurements sampled at 50 Hz, where the ground truth change points were independently annotated using video and sensor data. We selected 25 instances covering 17 indoor and 8 outdoor activities for various numbers of segments ranging from 1 to 6. We selected

Table 10: Runtime comparison of SWCPD and OT-based CPD methods

(a) Average runtimes and AUC scores for OT-baseline methods

| Method | Runtime (s) | AUC |
|---|---|---|
| OT-CPD | $425 \pm 150$ | $0.95 \pm 0.05$ |
| e-divisive | $5.9 \pm 3.1$ | $0.96 \pm 0.05$ |

(b) Average runtimes and AUC scores of SWCPD for different numbers of projections $L$

| $L$ | Runtime (s) | AUC | vs. OT-CPD | vs. e-divisive |
|---|---|---|---|---|
| 100 | $1.02 \pm 0.2$ | $0.87 \pm 0.1$ | $+41,979\%$ | $+478\%$ |
| 500 | $2.81 \pm 0.6$ | $0.95 \pm 0.1$ | $+15,024\%$ | $+109\%$ |
| 1000 | $3.33 \pm 0.74$ | $0.95 \pm 0.1$ | $+12,662\%$ | $+77\%$ |
| 5000 | $6.21 \pm 1.3$ | $0.97 \pm 0.07$ | $+6,743\%$ | $-5\%$ |

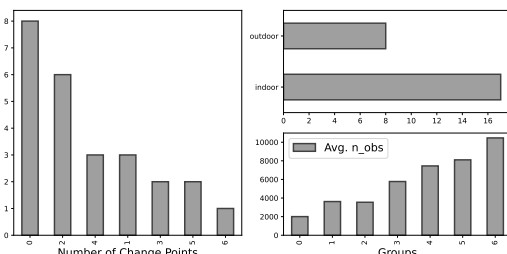

Figure 15: Summary of the data used for the change point detection experiments of HASC dataset.

8 instances with one segment, thus zero change points to asses the sensitivity and robustness of each method when the unknown underlying distribution does not change over time. Furthermore, we see that the average number of observations increases with more segments in the selected data see fig. 15. We specifically considered instances with a single segment to assess each method's robustness to false positives. Figure 16 illustrates the time series of an outdoor activity of a person. In this case, the person is performing three different stretches (standing adductor left, squat stretch for adductors, hamstring stretch right) Figure 17 shows AUC scores of our proposed method and baseline methods for five different annotation margins $\tau \in [25, 50, 100, 150, 200]$, such that if the annotated change point is at least $\tau$ instances away, it is classified as true positive thus contribution to the AUC score. We see that SWCPD shows superior AUC scores for any $\tau$, see Figure 17.

### C.2.4 OCCUPANCY

WCPD is based on the Sliced Wasserstein distance which is a metric from Optimal Transport (OT). To contextualize the computational performance of our proposed method for other OT-based detection methods such as OT-CPD, and e-divisive, we report the average wall-clock time and standard deviation in Table 11.

Table 11: Runtime comparison of SWCPD and OT-based CPD methods

(a) Average runtimes and AUC scores for OT-baseline methods

| Method | Runtime (s) | AUC |
|---|---|---|
| OT-CPD | $96.2 \pm 0.23$ | $0.41 \pm 0.00$ |
| e-divisive | $175.3 \pm 0.19$ | $0.34 \pm 0.00$ |

(b) Average runtimes and AUC scores of SWCPD for different numbers of projections $L$

| $L$ | Runtime (s) | AUC | vs. OT-CPD | vs. e-divisive |
|---|---|---|---|---|
| 100 | $28.2 \pm 0.8$ | $0.48 \pm 0.0$ | $+241\%$ | $+519\%$ |
| 500 | $59.4 \pm 1.25$ | $0.58 \pm 0.0$ | $+62\%$ | $+195\%$ |
| 1000 | $66.6 \pm 1.55$ | $0.59 \pm 0.0$ | $+45\%$ | $+163\%$ |

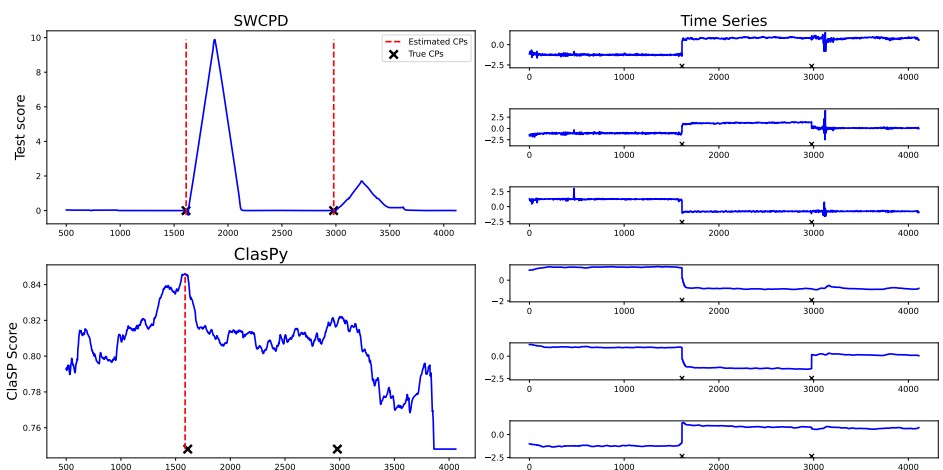

Figure 16: Comparison of Test scores obtained using SWCPD and ClaSP on subject number 243 (left hand side), and corresponding time series (right hand side).

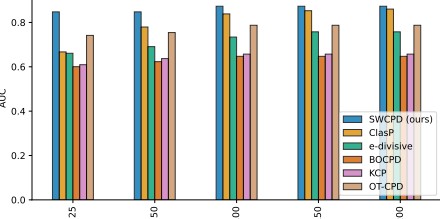

Figure 17: Shows average AUC scores for proposed method and baseline methods on the selected HAR data for different annotation margins $\tau$.

## D OMITTED PROOFS

**Lemma D.1.** *Let $X$ and $Y$ be two independent random variable such that $X \sim \Gamma(\alpha_1, \beta)$ and $Y \sim \Gamma(\alpha_2, \beta)$ with $\alpha_i, \beta \geq 0$ for $i = 1, 2$. Let $Z := X + Y$, then $Z \sim \Gamma(\alpha_1 + \alpha_2, \beta)$*

*Proof.* We consider independent Gamma random variables $X_k$ with different shape parameters $\alpha_k$ and fixed rate parameters $\beta$ for $k \in [N]$. Given the probability density function of $X_k$,

$$f_{X_k}(x) = \frac{\beta^{\alpha_k}}{\Gamma(\alpha_k)} x^{\alpha_k - 1} \exp(-\beta x),$$

we have the characteristic function

$$\varphi_{X_k}(t) = \mathbb{E}[e^{it X_k}] = \frac{\beta^{\alpha_k}}{\Gamma(\alpha_k)} \int_0^\infty x^{\alpha_k - 1} e^{-(\beta - it)x} \mathrm{d}x$$

$$= \left(1 - \frac{it}{\beta}\right)^{-\alpha_k}$$

for $k = 1, 2$ and $\alpha_1, \alpha 2 \geq 0$. Finally, we denote $Z = \sum_{k=1}^N X_k$ and $\underline{\alpha} = \sum_{k=1}^N \alpha_k$ and use

$$\varphi_Z(t) = \prod_{k=1}^N \varphi_{X_k}(t) = \left(1 - \frac{it}{\beta}\right)^{-\underline{\alpha}}.$$

$\square$

**Lemma D.2.** *Let $X \sim \mathcal{N}(0, \sigma^2)$, then $|X|^2 \sim \Gamma(\frac{1}{2}, \frac{1}{2\sigma^2})$ follows a Gamma distribution with shape parameter $\alpha = \frac{1}{2}$ and rate parameter $\beta = \frac{1}{2\sigma^2}$.*

*Proof.* We will first show that $|X|$ follows a half-normal distribution with scale $\sigma$. By definition, the probability density function of $X$ is $f_X(x) = \frac{1}{\sqrt{2\pi}\sigma} \exp\left(-\frac{x^2}{2\sigma^2}\right)$. Let us define $Y = |X|$, then each realization of $Y$ denoted as $y \in [0, \infty)$, such that,

$$F_Y(y) = \Pr(Y \leq y) = \Pr(|X| \leq y)$$

$$= \Pr(-y \leq X \leq y) = 2 \cdot \Pr(0 \leq X \leq y)$$

$$= 2 \cdot \int_0^y \frac{1}{\sqrt{2\pi}\sigma} \exp\left(-\frac{x}{2\sigma^2}\right) \mathrm{d}x$$

Finally, we obtain

$$f_Y(y) = \frac{\mathrm{d}}{\mathrm{d}y} F_Y(y) = \sqrt{\frac{2}{\pi}} \sigma^{-1} \exp\left(-\frac{y^2}{2\sigma^2}\right), \quad \text{for } y \geq 0,$$

which concludes that $Y = |X|$ follows a half normal distribution. Similar, we set $Z := Y^2$ and have,

$$F_Z(z) = \Pr(Z \leq z) = \Pr(Y \leq \sqrt{z}) = F_Y(\sqrt{z}),$$

since $Y \geq 0$. Subsequently, differentiating the CDF $F_Z(z)$ w.r.t. $z$ and using $\Gamma(\frac{1}{2}) = \sqrt{\pi}$, we obtain the following probability density function,

$$f_Z(z) = \frac{1}{2\sqrt{z}} f_Y(\sqrt{z}) = \frac{1}{\sqrt{2\pi z}\sigma} \exp\left(-\frac{z}{2\sigma^2}\right)$$

$$= \frac{1}{\Gamma(\frac{1}{2})\sqrt{2\sigma^2 z}} \exp\left(-\frac{z}{2\sigma^2}\right) \quad \text{for } z \geq 0.$$

which concludes the claim $|X|^2 \sim \Gamma(\frac{1}{2}, \frac{1}{2\sigma^2})$. $\square$

**Theorem D.3.** *[Berry-Esseen Berry (1941); Jacod & Protter (2012)] Let $(X_j)_{j>0}$ be an i.i.d. sequence of random variables with $\mathbb{E}[X_j] = 0, \mathbb{E}[X_j^2] = \sigma^2$, and finite third moments $\mathbb{E}[|X_j|^3] < \infty$, if we set $S_n = \frac{X_1 + X_2 + \cdots + X_n}{\sigma \sqrt{n}}$, then there exists a positive constant $C$ such that*

$$\sup_{t \in \mathbb{R}} |\mathbb{P}(S_n \leq t) - \Phi(t)| \leq C \frac{\mathbb{E}[|X_j|^3]}{\sigma^3 \sqrt{n}},$$

*where $\Phi(t)$ denotes the cdf of a standard normal distribution.*

*Proof.* We refer the reader to Jacod & Protter (2012). □

**Lemma D.4.** *Let $\theta \sim \mathcal{U}(\mathbb{S}^{d-1})$ and $\Sigma \in \mathbb{R}^{d \times d}$ p.s.d., then $\mathcal{Q} = \theta^T \Sigma \theta \xrightarrow{d} \mathcal{N}\left(\frac{tr(\Sigma)}{d}, \frac{2tr(\Sigma^2)}{d^2}\right)$.*

*Proof.* Let $x \sim \mathcal{N}(0, \mathbf{I}_d)$, we set $\theta = \frac{x}{||x||}$ such that the quadratic form $\mathcal{Q} = \theta^T \Sigma \theta = \frac{x^T \Sigma x}{||x||^2}$. We write

$$x^T \Sigma x = \sum_{i=1}^{d} \lambda_i x_i^2,$$

where $\lambda_1, \ldots, \lambda_d$ are the eigenvalues obtained after diagonalizing $\Sigma = U \Lambda U^T$. Let us set

$$S_d = \sum_{i=1}^{d} \lambda_i (x_i^2 - 1),$$

such that $\mathcal{Q} = \sum_{i=1}^{d} \lambda_i + S_d = \text{tr}(\Sigma) + S_d$, where $S_d$ is a sum of independent random variables. We apply $Theorem\ D.3$ where $X_i = \lambda_i(x_i^2 - 1)$, $\text{Var}(X_i) = 2\lambda_i^2$, and $\mathbb{E}[|X_i^3| = \lambda_i^3 \mathbb{E}[|x_i^2 - 1|^3] = \lambda_i^3 c$, then we have

$$\sup_{t \in \mathbb{R}} |\mathbb{P}(S_n \leq t) - \Phi(t)| \leq C \frac{\sum_i \lambda_i^3}{\left(\sum_i \lambda_i^2\right)^{\frac{3}{2}}}$$

which gives a uniform bound of the differences between the distribution of the random projections and a standard normal distribution which is dependent on the spectrum of $\Sigma$. Moreover, since $\mathbb{E}[||x||^2] = d$, and $\text{Var}(||x||^2) = 2d$, we have $\mathcal{Q} = \theta^T \Sigma \theta = \frac{\text{tr}(\Sigma)}{d} + \mathcal{N}(0, \frac{\sigma^2}{d^2}) = \mathcal{N}(\frac{\text{tr}(\Sigma)}{d}, \frac{2\text{tr}(\Sigma^2)}{d^2})$. Such that

$$\mathcal{Q} \xrightarrow{d} \mathcal{N}\left(\frac{\text{tr}(\Sigma)}{d}, \frac{2\text{tr}(\Sigma^2)}{d^2}\right)$$

□

**Theorem D.5.** *Let $\mathbb{P}, \mathbb{Q}$ denote two probability distributions on $\mathbb{R}^d$ with finite p'th moments then $w_2^2(\theta)[\mathbb{P}^\theta, \mathbb{Q}^\theta] \sim \Gamma$ as $d \to \infty$.*

*Proof.* We denote the probability distribution of $X, Y$ with $\mathbb{P}, \mathbb{Q}$ respectively. We write $Z = \langle X, \theta \rangle$, $W = \langle Y, \theta \rangle$ modeling the projections $T_\#^\theta \mathbb{P}, T_\#^\theta \mathbb{Q}$. First, we consider the projection for a specific sample $x_i$ denoted $z_i = \langle x_i, \theta \rangle$. Thus, for a fixed sample, we have

$$\mathbb{E}[z_i] = \mathbb{E}[\langle x_i, \theta \rangle] = \sum_{k=1}^{d} x_{ik} \mathbb{E}[\theta_k] = 0,$$

$$\text{Var}(z_i) = \mathbb{E}[z_i^2] - \mathbb{E}[z_i]^2 = \sum_{k=1}^{d} x_{ik}^2 \mathbb{E}[\theta_k^2] = \frac{1}{d} ||x_i||^2$$

leading to $z_i \sim \mathcal{N}(0, \frac{1}{d}||x_i||^2)$ for large $d$.

Now, we fix some projection direction $\theta_l \sim \mathcal{U}(S^{d-1})$ and consider a sample set $X = (x_1, x_2, \ldots, x_n)$, we set $z_l = \langle X, \theta_l \rangle$, then,

$$\mathbb{E}[z_l] = \sum_{k=1}^{d} \mathbb{E}[X_{ik}] \theta_{lk},$$

$$\text{Var}(z_l) = \mathbb{E}[z_l^2] - \mathbb{E}[z_l]^2 = \sum_{k=1}^{d} \mathbb{E}[X_k^2] \theta_{lk}^2 + 2 \sum_{k,m=1}^{d} \mathbb{E}[X_k X_m] \theta_{lk} \theta_{lm} - \mathbb{E}[z_l]^2$$

$$= \sum_{k=1}^{d} \mathbb{E}[X_k^2] \theta_{lk}^2 - \sum_{k=1}^{d} \mathbb{E}[X_k]^2 \theta_{lk}^2 + 2 \sum_{k,m=1}^{d} \mathbb{E}[X_k X_m] \theta_{lk} \theta_{lm} - \mathbb{E}[X_k] \mathbb{E}[X_m] \theta_{lk} \theta_{lm}$$

after rearranging the terms, we have $\text{Var}(z_l) = \theta_l \Sigma_X \theta_l^T$, such that $z_l \sim \mathcal{N}\left(\sum_{k=1}^d \mathbb{E}[X_{ik}]\theta_{lk}, \theta_l \Sigma_X \theta_l^T\right)$. Analogously, we consider a sample set $Y = (y_1, y_2, \ldots, y_n)$ and write $w_l := \langle Y, \theta_l \rangle$, subsequently, we see $w_l \sim \mathcal{N}\left(\sum_{k=1}^d \mathbb{E}[Y_{ik}]\theta_{lk}, \theta_l \Sigma_Y \theta_l^T\right)$.

The main step in the calculation of the Sliced Wasserstein distance is the utilization of the closed expression of the Wasserstein distance between two univariate distributions, which reads that for two probability distributions with $p$ finite moments, the Wasserstein distance boils down to

$$\mathrm{W}_p^p(\mathbb{P}, \mathbb{Q}) = \int_0^1 |F_\mathbb{P}^{-1}(u) - F_\mathbb{Q}^{-1}(u)|^p \mathrm{d}u, \tag{7}$$

where $F^{-1}$ denote the inverse CDF of $\mathbb{P}, \mathbb{Q}$ indicated by the subscript. Note, if we plug in $z_l, w_l$ for $\mathbb{P}$ and $\mathbb{Q}$ in eq. (7), we obtain the $p$ Wasserstein distance for the projection direction $\theta_l$. Since we derived that the distributions for a fixed projection behave Gaussian, we consider

$$F_{z_l}^{-1}(u) = \sqrt{2\theta_l \Sigma_X \theta_l^T} \cdot \mathrm{erf}^{-1}(2u - 1) + \mu_{z_l}$$

where $\mathrm{erf}^{-1}$ denotes the inverse of the Gauss error function. We have $D(u) := F_{z_l}^{-1}(u) - F_{w_l}^{-1}(u)$,

$$D(u) = \left(\sqrt{2\theta_l \Sigma_X \theta_l^T} - \sqrt{2\theta_l \Sigma_Y \theta_l^T}\right) \cdot \mathrm{erf}^{-1}(2u - 1) + \mu_{z_l} - \mu_{w_l}.$$

Let us fix $u$ and consider all possible projections $\theta$, we see $\mathbb{E}_\theta[\theta \Sigma \theta^2] = \frac{1}{d}\mathrm{tr}(\Sigma)$, while $\mathbb{E}[\mu_z] = \mathbb{E}[\mu_w] = 0$, therefore $\mathbb{E}[D(u)] = \left(\sqrt{\frac{2}{d}\mathrm{tr}(\Sigma_X)} - \sqrt{\frac{2}{d}\mathrm{tr}(\Sigma_Y)}\right) \cdot \mathrm{erf}^{-1}(2u - 1)$, with Theorem D.4 we have $\text{Var}(\theta^T \Sigma \theta) = \frac{2\mathrm{tr}(\Sigma^2)}{d}$ for large $d$. Thus $\sigma_u^2 = \text{Var}(D(u)) = \mathrm{erf}^{-1}(2u - 1)^2 \text{Var}\left(\sqrt{2\theta_l \Sigma_X \theta_l^T} - \sqrt{2\theta_l \Sigma_Y \theta_l^T}\right)$ which is convex in u. This means that the variance increases in the tails. For each $u$ the differences of the inverse CDF are Gaussian for large $d$ with similar variance $\sigma_u^2$ with $D(u) \sim \mathcal{N}(\mu_u, \sigma_u^2)$. Therefore, $|D(u)|^2 \sim \chi_1^2(\lambda_u)$, note that the mean has a fixed value scaled by the error function, such that we can factor this term out. Normalizing the random variables will lead to a sum of Gamma random variables Lemma $D.2$ which is also Gamma distributed Lemma $D.1$, however the exact shape and rate parameter are not directly obtainable as approximation with the normalization is applied. □

*Proof of Proposition 3.2.* Suppose, we have i.i.d. samples $x_1, \ldots, x_n \sim \Gamma(\alpha, \beta)$ which we denote as $X_n$. For a Gamma distribution with shape $\alpha$ and rate $\beta$, we have $\mu = \frac{\alpha}{\beta}$ and $\sigma^2 = \frac{\alpha}{\beta^2}$. We write $\overline{X}_n = \frac{1}{n}\sum_{i=1}^n x_i$ for the sample mean and $S_n^2 = \frac{1}{n-1}\sum_{i=1}^n (x_i - \overline{X}_n)^2$ for the sample variance. Then, we have the following Method of Moment estimates for $\alpha$ and $\beta$

$$\widehat{\alpha} = \frac{\overline{X}_n^2}{S_n^2}, \quad \widehat{\beta} = \frac{\overline{X}_n}{S_n^2}.$$

By the Central Limit Theorem, we know that for large $n$, the sample mean and variance converges to a normal distribution, with

$$\sqrt{n}\left(\widehat{\alpha}\widehat{\beta}^{-1} - \mu\right) \xrightarrow{d} \mathcal{N}\left(0, \sigma^2\right)$$

$$\sqrt{n}\left(S_n^2 - \sigma^2\right) \xrightarrow{d} \mathcal{N}\left(0, \text{Var}(S_n^2)\right)$$

where, with *Theorem 1* from Cho & Cho (2008), $\text{Var}(S_n^2) \approx n^{-1}(3\sigma^2 + 2\sigma^2\mu2 - \sigma^4) = \frac{2\alpha^2}{n\beta^4}$ for $n \to \infty$. We use the asymptotic normality of sample mean and variance and apply the delta method to derive an approximation of the variance of $\hat{\alpha}, \hat{\beta}$. For a smooth differentiable function $g(\theta)$ and a sequence of random variables $\theta_n$, if $\sqrt{n}(\theta_n - \theta) \xrightarrow{d} \mathcal{N}(0, \Sigma)$, then $\sqrt{(n)}(g(\theta_n) - g(\theta)) \xrightarrow{d} \mathcal{N}\left(0, \nabla g(\theta)^T \Sigma \nabla g(\theta)\right)$. Beginning with the estimate for $\alpha$, we set

$$g(\overline{X}_n, S_n^2) = \frac{\overline{X}_n^2}{S_n^2},$$

with

$$\nabla g \left( \overline{X}_n^2, S_n^2 \right)^T = \left( 2 \frac{\overline{X}_n}{S_n^2}, -\frac{\overline{X}_n^2}{(S_n^2)^2} \right).$$

The covariance matrix $\Sigma$ consists of $\mathrm{Var}(\overline{X}_n)$ and $\mathrm{Var}(S_n^2)$ on the diagonal and $0$ on the off diagonal elements due to the fact that for large $n$ sample mean and variance are uncorrelated. Therefore, we have

$$\mathrm{Var}(\hat{\alpha}) \approx \left( \frac{2\overline{X}_n}{S_n^2} \right)^2 \cdot \mathrm{Var}(\overline{X}_n) + \left( \frac{\overline{X}_n^2}{(S_n^2)^2} \right)^2 \cdot \mathrm{Var}(S_n^2),$$

and plugging the estimator for sample mean and variance in, we may simplify the expression to

$$\mathrm{Var}(\hat{\alpha}) \approx \frac{4\alpha^2}{n} + \beta^4 \cdot \mathrm{Var}(S_n^2) = \frac{6\alpha^2}{n}.$$

For the estimator of $\beta$, we set

$$g(\overline{X}_n, S_n^2) = \frac{\overline{X}_n}{S_n^2},$$

repeating the steps from above leads to,

$$\mathrm{Var}(\hat{\beta}) \approx \left( \frac{1}{S_n^2} \right)^2 \cdot \mathrm{Var}(\overline{X}_n) + \left( \frac{\overline{X}_n}{(S_n^2)^2} \right)^2 \cdot \mathrm{Var}(S_n^2),$$

which we simplify to

$$\mathrm{Var}(\hat{\beta}) \approx \frac{\beta^2}{n \cdot \alpha} + \frac{\beta^6}{\alpha^2} \cdot \mathrm{Var}(S_n^2).$$

$\square$