# OpenReview forum: "High-Dimensional Online Change Point Detection with Adaptive Thresholding and Interpretability"
_ICLR.cc/2026/Conference — Submitted to ICLR 2026_

### Official Review · Reviewer_EQ8h · 2025-11-01

**Soundness:** 2
**Presentation:** 3
**Contribution:** 2
**Rating:** 4
**Confidence:** 3

**Summary:**

The authors consider the problem of nonparametric online change point detection. They suggest an algorithm based on sliced Wasserstein distance. Because of the reduction to one-dimensional projections, the authors significantly reduce computational expenses. Moreover, they establish (see Theorem 3.1) that the Wasserstein distance between random projections tends to the gamma distribution as the dimension increases. This suggest an approach for automatic threshold choice based on asymptotic distribution of the sliced Wasserstein distance. The authors illustrate the performance of their algorithm on synthetic and real-world data.

**Strengths:**

Theorem 3.1 about asymptotic distribution of the sliced Wasserstein distance (as $d \rightarrow \infty$). allows for adaptive choice of the threshold.

**Weaknesses:**

1. While Proposition 3.2 gives an intuition about running length of the detection procedure, its detection delay remains unexplored. This leaves open the question of optimality of the suggested approach.

2. Some important references are missing. A variant of CUSUM was also studied in [Yu et al., 2023]. Methods based on different f-divergences and density-ratio estimation were considered in [Liu et al., 2013] and [Hushchyn, A. Ustyuzhanin, 2021]. Their change-point detection methods are based on KLIEP [Sugiyama, 2008], uLSIF [Kanamori et al., 2009], and RuLSIF [Yamada et al., 2013]. Approaches based on deep neural networks (with a statistics similar to the one used in GANs) were suggested in [Hushchyn et al., 2020], [Puchkin, Shcherbakova, 2023]. Finally, in [Cao et al., 2018] and [Markovich, Puchkin, 2024] the authors proposed algorithms based on online convex optimization and prediction with expert advice. I would suggest the authors to incorporate these references into the literature review during revision.

[Sugiyama, 2008] M. Sugiyama, T. Suzuki, S. Nakajima, H. Kashima, P. von Bunau, and M. Kawanabe. Direct importance estimation for covariate shift adaptation. Annals of the Institute of Statistical Mathematics, 60(4):699–
746, 2008.

[Kanamori et al., 2009] T. Kanamori, S. Hido, and M. Sugiyama. A least-squares approach to direct importance estimation. Journal of Machine Learning Research, 10:1391–1445, 2009.

[Yamada et al., 2013] M. Yamada, T. Suzuki, T. Kanamori, H. Hachiya, and M. Sugiyama. Relative density-ratio estimation for robust distribution comparison. Neural Computation, 25(5):1324–1370, 2013.

[Liu et al., 2013] S. Liu, M. Yamada, N. Collier, and M. Sugiyama. Change-point detection in time-series data by relative density-ratio estimation. Neural Networks, 43:72–83, 2013.

[Cao et al., 2018] Y. Cao, L. Xie, Y. Xie, and H. Xu. Sequential change-point detection via online convex optimization. Entropy, 20(2):108, 2018.

[Hushchyn et al., 2020] M. Hushchyn, K. Arzymatov, and D. Derkach. Online neural networks for change-point detection. Preprint, arXiv:2010.01388, 2020.

[Hushchyn, A. Ustyuzhanin, 2021] M. Hushchyn and A. Ustyuzhanin. Generalization of change-point detection in time series data based on direct density ratio estimation. J. Comput. Sci., 53:Paper No. 101385, 8, 2021.

[Puchkin, Shcherbakova, 2023] N. Puchkin and V. Shcherbakova. A Contrastive Approach to Online Change Point Detection. In Proceedings of The 26th International Conference on Artificial Intelligence and Statistics, volume 206 of Proceedings of Machine Learning Research, pages 5686–5713, 2023.

[Yu et al., 2023] Y. Yu, O. H. M. Padilla, D. Wang, and A. Rinaldo. A note on online change point detection. Sequential Analysis, 42(4):438–471, 2023.

[Markovich, Puchkin, 2024] Score-based change point detection via tracking the best of infinitely many experts. Preprint, arXiv:2408.14073, 2024.

**Questions:**

1. According to my experience, the threshold choice based on asymptotic distribution of the test statistic may be not the best one (this agrees with one of limitations you mentioned in Section 5). For instance, the automatic threshold choice for the kernel-based test statistic [Li et al., Seq. Anal., 2019] lead to frequent false alarms. Has a similar problem occurred, for instance, on the Room Occupancy data (where the time series has only 4 components)? Could you elaborate on how you chose a threshold in that case?

2. Could you add a comparison of your algorithm with any of [Liu et al., 2013], [Hushchyn et al., 2020], [Markovich, Puchkin, 2024]? All these papers considered the human activity recognition data set. Besides, [Hushchyn et al., 2020] checked performance of their procedure on the MNIST data set, while [Markovich, Puchkin, 2024] applied their approach to the Room Occupancy data.

---

> ### Author Response · Authors · 2025-11-17
>
> We thank the reviewer for the thoughtful feedback.
>
> First, we want to comment on the weaknesses raised:
>
> 1) Proposition 3.2 is about confidence intervals for the MoM estimates of shape and rate parameters, not the running length. We empirically evaluate the detection delay by calculating the average detection delay across all experiments.
>
> 2) We thank the reviewer for pointing out further relevant literature. We will discuss it in the related work section in the revised version.
>
> Below, we want to answer your questions raised in your review.
>
> **Q1:** Thank you for this insightful observation. We also agree that the threshold selection based on asymptotic distributions can be suboptimal when there are limited observations or heavy-tailed distributions. This aligns with the limitations we discussed in Section 5. In light of your comment, we will highlight alternative adaptive thresholding strategies (e.g., permutation-based or bootstrapping) for future work in our revised manuscript.
>
> Generally, deriving a fixed threshold is difficult without specific assumptions on the underlying process. Our proposed method relies on an **adaptive threshold** which is derived from a Gamma distribution obtained by the moving average of the $m=\min(K_{\max},t)$ most recent previous estimates of shape and rate parameters. The threshold is the $p$-quantile of the estimated Gamma distribution, thus it also **controls** the **sensitivity to false positives**. The threshold is adaptive to temporal changes since it is a moving average estimate. Below, we illustrate the sensitivity of false positives to a change in $p$, and the length of the moving average window $K_{\max}$. Higher $p$-values yield more false positives as the adaptive threshold is decreased on average.
>
> |$K_{max}$|$p=0.99$|$p=0.95$|$p=0.90$|$p=0.80$|
> |---|---|---|---|---|
> |$50$|$2$|$5$|$8$|$13$|
> |$100$|$5$|$7$|$11$|$15$|
> |$200$|$4$|$7$|$7$|$14$|
> |$250$|$5$|$6$|$6$|$13$|
>
> We fix the random seed across the experiments.
>
> Data: Occupancy, $w=250$, $L=1000$, Wasserstein_order $=2$.
>
> **Q2:** We thank the reviewer for pointing out related detection methods that consider similar datasets to our study. We now compare our method against three Deep learning-based approaches proposed in [1] and [2], and a baseline change point detection Rulif [3] on MNIST as proposed in [1]. Note in [1] they use a downsampled version with $8\times8$ pixels while we consider the full $28\times28$ spectrum. Additionally, in [1] the results reported use a margin of 50 samples $\tau=50$, while below, we report metrics based on smaller margins $\tau=20,30$. Reported results are averaged over 10 sequences from [1], and hyperparameters are selected accordingly. For SWCPD we set $w=K_{\max}=100$, wasserstein order $=2$ and $\alpha=1-p=0.075$, $L=1000$.
>
> |Method|AUC||FP|||||
> |---|---|---|---|---|---|---|--|
> ||$\tau=20$| $\tau=30$|$\tau=20$|$\tau=30$| **Cov**|**DD**|**Runtime (s)**|
> |SWCPD|$0.97 \pm 0.04$|$0.99 \pm 0.02$|$0.2$ (0;1)|$0.0$ (0;0)|$0.90 \pm 0.01$|$11.1$ (9.3;12.4)|$4.25 \pm 0.1$|
> |DeepClf [1]|$0.86 \pm 0.09$|$0.96 \pm 0.08$|$1.5$ (0;3)|$0.9$ (0;2)|$0.89 \pm 0.02$|$10.7$ (5.6;17)|$7.64 \pm 3.26$|
> |DeepRulif [1]|$0.80 \pm 0.09$|$0.90 \pm 0.05$|$2.2$ (0;4)|$1.2$ (0;3)|$0.88 \pm 0.02$|$12.1$ (7.1;18)|$17.1 \pm 0.2$|
> |DeepNN [2]|$0.24 \pm 0.11$|$0.96 \pm 0.01$|$8.3$ (6;10)|$0.0$ (0;0)|$0.80 \pm 0.02$|$22.1$ (21.4;23.5)|$284 \pm 1.3$|
> |Rulif [3]|$0.72 \pm 0.09$|$0.84 \pm 0.1$|$1.6$ (0;3)|$0.4$ (0;1)|$0.66 \pm 0.02$|$11.5$ (6.8;14.3)|$1.4 \pm 0.02$|
>
>
> The high $p$ value contributes to the low false positives.
>
> We hope this clarifies your question and highlights that our proposed method achieves competitive or favourable detection performance against Deep learning based methods with minimal false positives and runtime advantages.
>
> References:
>
> [1] M. Hushchyn, K. Arzymatov, and D. Derkach. Online neural networks for change-point detection. Preprint, arXiv:2010.01388, 2020.
>
> [2] Puchkin, N., & Shcherbakova, V. (2023, April). A contrastive approach to online change point detection. In International Conference on Artificial Intelligence and Statistics (pp. 5686-5713). PMLR.
>
> [3] S. Liu, M. Yamada, N. Collier, and M. Sugiyama. Change-point detection in time-series data by relative density-ratio estimation. Neural Networks, 43:72–83, 2013.

---

> > ### Comment · Reviewer_EQ8h · 2025-11-27
> >
> > I would like to thank the authors for their response. They have adequately addressed my questions, and I have no further concerns. At this moment, I am tending to raise my score to 6.

---

### Official Review · Reviewer_7mTU · 2025-11-01

**Soundness:** 3
**Presentation:** 3
**Contribution:** 3
**Rating:** 6
**Confidence:** 5

**Summary:**

This paper introduces Sliced Wasserstein Change Point Detection (SWCPD), a novel framework for change point detection (CPD) in high-dimensional data.

The cornerstone of this framework is a new insight: the theoretical result showing that random Sliced Wasserstein slices follow a Gamma distribution for adaptive thresholding. This finding is leveraged by the authors practically for CPD. Multivariate data is transformed into univariate signals using Sliced Wasserstein (SW) distance and then a hierarchical procedure for generating contrastive explanations of detected changes is implemented.

The method is evaluated on synthetic and real-world datasets (MNIST, Human Activity Recognition, Occupancy) and compared against established baselines (BOCPD, e-divisive, KCP, OT-CPD, ClaSP).

**Strengths:**

- Novel theoretical characterization connecting SW projections to Gamma distributions, enabling principled statistical hypothesis testing for CPD.

- First method to combine Sliced Wasserstein distance with adaptive thresholding and contrastive explanations in a unified framework.

- Theorem 3.1, Propositions 3.2, and supporting lemmas are seemingly sound mathematical foundations

- Comprehensive experimental evaluation including synthetic and real datasets, with ablation studies and parameter sensitivity analysis

- Reproducible methodology with clearly described experimental settings and baselines.
- Strong empirical results, particularly in reducing false positives

- Figures and tables generally well-designed; boxplots in Figure 10 effectively communicate parameter sensitivity.

- Relevant to high-stakes domains (finance, healthcare, autonomous systems) where both accuracy and interpretability are critical.

- SWCPD has computational efficiency, which is meaningful in practice for streaming applications.

**Weaknesses:**

Theoretical Limitations
- Assumption Gap (in Section 3.1): Theorem 3.1 requires d → ∞ for the Gamma distribution approximation, but practical datasets have moderate dimensions. While Table 5 shows empirical validation works for d ≥ 20, the paper would benefit from:
- Providing finite-sample bounds or approximation error rates for moderate d
- Specifying minimum dimension requirements for reliable performance
- Discussing behavior when theoretical assumptions are violated
- Temporal Correlation (in Section 3.2): The moving average smoothing (step 2 of the algorithm) applies to temporally correlated sliding windows, yet the paper assumes i.i.d. random projections for statistical validity. The paper acknowledges this, but dismisses it without rigorous justification.
- Method of Moments Limitations: MoM estimation is sensitive to outliers and heavy-tailed distributions.

The paper should:
- Compare MoM to maximum likelihood estimation empirically
- Analyze robustness to distribution misspecification
- Provide diagnostic criteria for when the Gamma approximation fails

Hyperparameter Selection (Section 4.2):
- Tolerance criterion (Algorithm 2, line 3): The stopping criterion β̂ ≤ tol lacks justification. What is an appropriate tolerance value? How sensitive are explanations to this choice?
- Window length w: Figure 3 shows high sensitivity to w, but no principled selection strategy is provided beyond grid search. Next steps should develop data-driven selection methods (e.g., cross-validation, information criteria).
- Wasserstein order p: The choice between p=2,4,6 appears dataset-dependent (Table 2, Figure 3) with no clear guidance. Provide decision rules based on drift characteristics.

Experimental Concerns
- Limited Dataset Diversity: Four datasets (two relatively simple: MNIST, Occupancy; one tied: HAR with 0.59 e-divisive).
- Missing challenging scenarios: high-noise environments, gradual drifts, multimodal distributions, very high dimensions (d > 100).
- Financial time series and sensor data mentioned in introduction but not evaluated.

Statistical Rigor:
- Confidence intervals on metrics would strengthen claims (only AUC has std in Table 3).
- No runtime comparisons with non-OT baselines (Table 8-9 only compare OT methods).
With L=5000 projections, practical speed claims need verification for streaming applications with strict latency requirements.

**Questions:**

To strengthen the background around Wasserstein-based change detection, it would be helpful to frame the discussion in consideration of prior OT/WD approaches, including WATCH and its lifelong extension (LIFEWATCH), which perform unsupervised CPD by monitoring Wasserstein shifts in high-dimensional streams. Beyond these, Cheng et al. (ICASSP’20) proposed an optimal transport CPD and segment-clustering pipeline grounded in the Wasserstein two-sample test. From a statistical perspective, Horváth et al. (Annals of Statistics, 2021) analyze sequential monitoring via a weighted Wasserstein distance with asymptotic and finite-sample guarantees, which I believe is a useful context for your hypothesis-testing claims.

Can you provide finite-sample error bounds or convergence rates for the Gamma approximation in Theorem 3.1? Specifically, what is the approximation error for moderate dimensions (d = 10-50) commonly seen in practice?

What is the practical minimum dimension d for reliable performance? Table 5 shows empirical validation, but can you provide theoretical guidance or diagnostic criteria for when the method should or shouldn't be applied?

You acknowledge temporal correlation (in line 218) but dismiss concerns without rigorous justification. Can you provide theoretical analysis or simulation studies quantifying the impact on confidence interval validity?

Can you show empirical evidence of robustness across different correlation structures? Can you provide guidelines for adjusting parameters when correlation is strong?

How should practitioners set the tolerance parameter “tol” in Algorithm 2? Can you provide a sensitivity analysis or specify the selection criteria?

Several improvements appear modest given the reported uncertainty. For example, HAR shows 0.85 ± 0.12 (yours) vs. 0.84 ± 0.15 (ClaSP), and Occupancy shows 0.59 vs. 0.52-0.58 (baselines). Can you provide statistical significance tests (such as paired t-tests or Wilcoxon tests) to confirm these improvements are not due to random variation?

---

> ### Author Response · Authors · 2025-11-20
> **Answer (1/2)**
>
> We thank the reviewer for the thoughtful review and insightful feedback.
>
> First, we want to comment on some weaknesses mentioned.
>
> **W1 (Theoretical Limitations):** We acknowledge that MLE has some advantages over MoM in general. However, MLE for a Gamma distribution has no closed-form solution. It requires numerical approximation, which can be slow and is not applicable for streaming data [1]. Therefore, we rely on MoM, which has a closed-form solution. Table 5 provides insights into the minimal dimension required and how the number of projections can relax it.
>
>
> **W2 (Experimental \& Statistical Evaluation):** Below, we add a justified stopping criterion (**Q5**). We have high-noise environments for the synthetic data. The marginal distribution of a variable with abrupt mean shift is multimodal by definiton, therefore each data can be regarded as a process of $d$ multimodal distributions. HAR is sensor data. We report **std** for **AUC**, **COV**, and **min max** ranges for **FP** and **DD**. Theoretical runtime complexity as a function of  $L$, $d$, and $w$ is provided in Appendix C.2 Table 6. In addition, we include empirical runtime comparisons in Table 8 and Table 9 against other **OT-based methods** to **contextualize** the efficiency of SWCPD.
>
>
> **Q1:** The Sliced Wasserstein distances involve two key aspects (1) the empirical approximation of the underlying distribution and (2) the Monte Carlo estimate of the integral over projections. In [2], the authors separately bound the MC error and the sample error for which Prop. 5 gives an upper bound in terms of $n$,$d$,$L$, and covariance matrices.
>
> **Q2:** We include a study about Gaussian tendencies of the random projections using the Shapiro-Wilk test across different dimensions $d$ with $N=100$ samples and  $L$ projection counts. Table 5 shows that normality becomes valid for $d>20$ and in most cases for $d>10$.
>
> We follow your request and now report the Mean Absolute Error (MAE) between the **theoretical quantiles** and **observed quantiles**. The theoretical quantiles are derived from a Gamma distribution based on the MoM estimates from the random projections. The observed quantiles are calculated based on the empirical distribution of the random projections. For each dimension, we simulate two independent datastreams, each consisting of $d$ independent Gaussian distributions with a uniformly sampled mean. We vary $d$ and $L$, use fixed random seeds, and report the results for 10 trials below.
>
> | Dimension ($d$) | L=100 | L=1.000 | L=10.000 |
> | :---: | :---: | :---: | :---: |
> | 5 | $1.1226 \pm 0.1722$ | $0.3586 \pm 0.0326$ | $0.3751 \pm 0.0106$ |
> | 10 | $0.3479 \pm 0.0631$ | $0.3187 \pm 0.0638$ | $0.1499 \pm 0.0090$ |
> | 20 | $0.4014 \pm 0.0724$ | $0.1652 \pm 0.0373$ | $0.01347 \pm 0.0158$ |
> | 100 | $0.2912 \pm 0.0451$ | $0.1100 \pm 0.0154$ | $0.0470 \pm 0.0124$ |
> | 200 | $0.2987 \pm 0.0535$ | $0.1344 \pm 0.0485$ | $0.0448 \pm 0.0142$ |
>
> We suggest that practitioners should use a sufficient number of projections whenever possible, $L>1.000$.
>
> **Q3:** In Appendix B.2, we show the asymptotic behaviour of the confidence intervals.
>
> **Q4:** Here, we empirically show detection results for our proposed method and KCP for high noise environments with variance changes. For this, we simulate a Mixture distribution 50/50 (Gaussian/exponential distributions) with $d=100$, and a total of 1500 samples with four regimes. We select the same base variance for all feature distributions and randomly change the variance for 15 features in the range of $(2\sigma_{base},2\sigma_{base}+1)$. We see that while the AUC and Covering score are superior or competitive to the baseline method, we significantly **improve** the **false positives**, which underlines our claim. We set $w=K_{\max}=100,p=2,L=5000,q=0.95$, all results are averaged over five runs.
>
> |$\sigma_{base}$|range of $\sigma_{change}$|Method|AUC|Covering|FP|DD|
> |---|---|---|---|---|---|---|
> |$0.1$|$(0.2,1.2)$|SWCPD|$1.0 \pm 0.0$|$0.94 \pm 0.01$|$0.0$ (0,0)|$15.26 \pm 1.8$|
> |$0.1$|$(0.2,1.2)$|KCP|$0.55 \pm 0.01$|$0.65 \pm 0.04$|$28.6$ (18,34)|$107.21 \pm 17.95$|
> |$0.3$|$(0.6,1.6)$|SWCPD|$0.9 \pm 0.01$|$0.77 \pm 0.1$|$0.2$ (0,1)|$29.9 \pm 6.3$|
> |$0.3$|$(0.6,1.6)$|KCP|$0.79 \pm 0.1$|$0.89 \pm 0.04$|$2.6$ (0,5)|$51.9 \pm 32.67$|
> |$0.5$|$(1,2)$|SWCPD|$0.88\pm 0.0$|$0.69 \pm 0.01$|$0.0$ (0,0)|$35.2 \pm 4.9$|
> |$0.5$|$(1,2)$|KCP|$0.93 \pm 0.01$|$0.98 \pm 0.04$|$0.4$ (0,2)|$8.2 \pm 10.95$|

---

> ### Author Response · Authors · 2025-11-20
> **Answer (2/2)**
>
> **Q5:** We will update the stopping criteria: In Algorithm 2, we update the removed feature from $Y$ with samples $X$. Suppose, we have observations $X_1,\dots,X_{N} \sim P_X$ and $Y_{1},\dots,Y_{N} \sim P_Y$ Witouth any drifts, we have $P_X=P_Y$ with $$\mu=\mathbb{E}[X]=\mathbb{E}[Y]$$ $$\Sigma=\text{Cov}(X)=\text{Cov}(Y),$$ where $m_X=\frac{1}{N}\sum_{i}^{N}X_{i}$, and $m_Y=\frac{1}{N}\sum_{i=1}^{N}Y_{i}$ denote the sample means. We consider $$||D||=||m(X)-m(Y)||,$$ then $$\mathbb{E}[||D||] \leq \sqrt{\frac{2}{N}tr(\Sigma)}$$ We have $D\sim \mathcal{N}(0,\frac{2}{N}\Sigma)$, we can decompose $\Sigma=U \Lambda U^{T}$. Then with $Z=U^{T}D$, it follows $||D||^{2}=\sum_{i=1}^{d}\frac{2}{N}\lambda_{i} \chi_{1}^{2}$. Therefore, we have $\mathbb{E}||D||^{2}=\frac{2}{N}tr(\Sigma)$, applying Jensen inequality yields $$\mathbb{E}[||D||] \leq \sqrt{\frac{2}{N}tr{\Sigma}}$$
>
> **We illustrate the sensitivity of this stopping criteria here:** https://anonymous.4open.science/r/E3C4/README.md
>
> We update the stopping criterion, which is now given in terms of $d$, $N$, and $\Sigma$, and offers a more elegant solution to $\beta$. We hope this clarifies your question and addresses the weakness mentioned.
>
> **Q6:** We thank the reviewer for this insight. Below, we report the results of a paired t-test between the AUC scores from our proposed and baseline methods. Occupancy consists of a single time series, thus it is not possible to apply the test accordingly.
> HAR ($d=12$) [3] is the data collected in [3], and the HAR ($d=561$) [4] comes from [4].
>
> |Method|Occupancy||HAR [3] || HAR[4]||MNIST||
> |---|---|---|---|---|---|---|---|---|
> |e-divisive|-||$p=1\times 10^{-4}$|✔️|$p=4 \times 10^{-5}$|✔️|$p=0.806$|➖|
> |KCP|-||$p=6\times10^{-6}$|✔️|$p=0.06$|➖|$p=0.02$|✔️|
> |BOCPD|-||$p=3\times10^{-7}$|✔️|$p=4\times 10^{-7}$|✔️|$p=5 \times 10^{-6}$|✔️|
> |CLasp|-||$p=0.21$|➖|$p=3\times 10^{-10}$|✔️|$p=2 \times 10^{-10}$|✔️|
> |OT-CPD|-||$p=$|✔️|$p=3\times 10^{-10}$|✔️|$p=2 \times 10^{-10}$|✔️|
>
> We will emphasize that performance in terms of AUC is equal for some baseline methods and will bold values accordingly. Our main claim that our method significantly **reduces false positives** while maintaining competitive or superior detection performance still holds.
>
> References:
>
> [1] Minka, Thomas P. "Estimating a gamma distribution." Microsoft Research, Cambridge, UK, Tech. Rep (2002).
>
> [2] Sloan Nietert, Ziv Goldfeld, Ritwik Sadhu, and Kengo Kato. Statistical, robustness, and computational guarantees for sliced wasserstein distances. Advances in Neural Information Processing Systems
>
>
> [3] Ermshaus, Arik, et al. "Human activity segmentation challenge@ ECML/PKDD’23." International Workshop on Advanced Analytics and Learning on Temporal Data. Cham: Springer Nature Switzerland, 2023.
>
> [4] Reyes-Ortiz, Jorge, et al. "Human Activity Recognition Using Smartphones." UCI Machine Learning Repository, 2013, https://doi.org/10.24432/C54S4K.

---

### Official Review · Reviewer_rWAP · 2025-11-01

**Soundness:** 2
**Presentation:** 2
**Contribution:** 2
**Rating:** 2
**Confidence:** 3

**Summary:**

In this paper, the authors present a framework for online change point detection in high-dimensional data with Sliced Wasserstein distance. The idea is to transform multivariate data into one-dimensional projections, derives that random SW slices follow a Gamma distribution for hypothesis testing. An adaptive thresholding algorithm was proposed based on a significance level, and aimed to offer contrastive explanations via geometric properties of projections.

**Strengths:**

> The paper addresses scalability and explainability in high-dimensional CPD, which is a growing concern in sequential data analysis.

> It includes a derivation showing SW slices approximate a Gamma distribution, enabling some statistical testing.

> Experiments on synthetic and a few real datasets show minor reductions in false positives in certain setups.

> The adaptive threshold and projection-based explanations offer a straightforward way to add interpretability.

**Weaknesses:**

- While the paper evaluates on time series (e.g., HAR) and image-based streams (e.g., MNIST), it fails to compare against state-of-the-art CPD methods tailored to each modality. For time series, it misses deep learning-based SOTA like LSTM/Transformer CPD (e.g., recent kernel methods for multivariate series), relying instead on general baselines like CUSUM or OT-CPD. For images (MNIST as sequential data), it ignores computer vision-specific CPD techniques, such as video anomaly detection models (e.g., autoencoders or flow-based methods for change detection in image sequences), limiting claims of superiority.
  - A general methodology for fast online changepoint detection
  - DeepLocalization: Using change point detection for Temporal Action Localization
  - Adaptive Block-Based Change-Point Detection for Sparse Spatially Clustered Data with Applications in Remote Sensing Imaging

- Only a handful of datasets are used (synthetic, HAR, MNIST, Occupancy), with small sequences (e.g., 200 samples per class in MNIST). No broader testing on diverse high-stakes domains like finance (e.g., stock data) or cybersecurity (e.g., network traffic benchmarks), despite mentioned applications. Results may not generalize.

- Relies on i.i.d. assumptions in windows, which may not hold for correlated time series or image streams. Adaptive thresholding requires tuning (e.g., α, window size), and computational costs for many projections (L=5000) aren't quantified for real-time online use.
Underdeveloped Interpretability: Contrastive explanations via projections are basic; no user studies or quantitative metrics (e.g., fidelity) to validate if they truly aid understanding in practice.

- Theoretical insights (Gamma approximation) are asymptotic (d→∞), potentially weak for moderate dimensions. Overall, incremental over existing SW/OT-based CPD (e.g., Cheng et al., 2020b), without strong novelty.

**Questions:**

>> The paper lacks comparisons to modality-specific CPD SOTA. Could the authors explain the reason? I understand it has used HAR (time series) and MNIST (image streams), but baselines are general (e.g., CUSUM, OT-CPD). For time series, why not include deep CPD methods like Transformer-based detectors (e.g., from recent works on multivariate series)? For images, why omit CV-specific SOTA like optical flow or autoencoder-based video CPD? This omission weakens claims of "competitive or superior performance."

>> How generalizable are results beyond the limited datasets? Experiments focus on synthetic data, HAR, MNIST (as streams), and Occupancy, with small scales (e.g., 200 samples/class). Have you tested on larger, domain-specific benchmarks like financial time series (e.g., S&P 500 drifts) or cybersecurity traces (e.g., KDD Cup anomalies)? Without this, applicability to "high-stakes" fields seems overstated.

>> I wonder about the robust of the Gamma approximation in practice. Theorem 3.1 assumes d→$\infty$ for SW slices ~ Gamma, but datasets like HAR (d~561) or MNIST (d=784) are finite. Appendix shows p-values, but could you provide ablation on approximation error for lower d, or when distributions violate assumptions (e.g., non-Gaussian)? Moreover, the experiment is not comprehensive. It doesn't consider how the adaptive thresholding is sensitive to parameter choices. Algorithm depends on α, window size, lookback Kmax. Appendix grid search on MNIST/Occupancy shows variations, but no sensitivity analysis across all datasets. For low-drift synthetics, how does it avoid false positives/negatives compared to fixed thresholds in OT-CPD?

>> The discussion is not thorough. How about the computational trade-offs for online deployment? With L=5000 projections and window sizes w=50, online CPD might be slow for streaming data. Grid searches vary α, w, Kmax, but no runtime benchmarks vs. baselines. How does it scale for real-time apps (e.g., autonomous systems), and why no efficiency metrics?

>> I am confused about the validation of the interpretability of explanations? Projection-based contrastive explanations identify "discriminative features," but lack quantitative evaluation (e.g., explanation accuracy via perturbation tests) or qualitative user studies. On HAR/MNIST, do they align with domain knowledge (e.g., sensor features in activity shifts)? Could you add metrics like faithfulness? What about correlated data or non-i.i.d. windows? Windows assume independence, but time series (HAR) and image streams (MNIST) often have temporal correlations. This could bias SW distances or MoM estimates. Could you discuss mitigations, like decorrelation steps, or test on autocorrelated synthetics?

---

> ### Author Response · Authors · 2025-11-14
> **Answer (1/2)**
>
> We thank the reviewer for the thoughtful feedback.
>
> First, we want to comment on some of the weaknesses raised.
>
> **W1:** Our work focuses on a principled online change-point detection framework that makes minimal distributional assumptions and can be deployed without any training. Because deep-learning–based detectors typically require substantial training data and computational resources, we viewed them as belonging to a different class of methods and therefore outside the primary scope of this paper. For this reason, we did not include them in our comparison. We will clarify this distinction more explicitly in the manuscript to avoid confusion.
>
> **W2:** Below, we include an example for longer sequences, which shows generalization to various variations of change point regimes. We limited the maximum length of MNIST sequences as runtime complexity scales quadratically with the number of observations for the baseline methods.
>
> **W3:** We don't rely on i.i.d. samples within the windows, however, the random projections are i.i.d. sampled by definition, such that the Wasserstein distances obtained for each random projection are independent.
>
> **W4:** The approximation also holds for moderate dimensions with an increasing number of random projections, see the table below (Q3).
>
> Below, we answer your questions raised in your initial review.
>
> **Q1:** This is related to **W1**, our porposed method is nonparametric and training-free. Deep learning-based methods require substantial training and hyperparameter tuning. They are less suitable for the purpose of our paper, but we acknowledge that such methods exist and may achieve higher performance once trained. We never claim superiority over those methods. Therefore, we deliberately claim competitive performance against popular (online/offline) CPD methods, which we demonstrate. Comparing our method against two Deep learning approaches proposed in [1] on MNIST (Note in [1] they use a downsampled version with 8x8 pixels) yields the following result, averaged over 10 MNIST sequences from [1]:
>
> |Method|AUC||FP||||
> |---|---|---|---|---|---|---|
> ||$\tau=20$| $\tau=30$|$\tau=20$|$\tau=30$| **Cov**|**DD**|
> |SWCDP|$0.92 \pm 0.04$|$0.94 \pm 0.02$|$0.2$ (0;1)|$0.0$ (0;0)|$0.90 \pm 0.01$|$11.1$ (9.3;12.4)|
> |DeepClf|$0.86 \pm 0.09$|$0.92 \pm 0.08$|$1.5$ (0;3)|$0.9$ (0;2)|$0.89 \pm 0.02$|$11.2$ (5.6;19.6)|
> |DeepRulif|$0.84 \pm 0.09$|$0.90 \pm 0.05$|$1.8$ (0;4)|$1.1$ (0;3)|$0.88 \pm 0.02$|$11.4$ (6.4;21.5)|
>
> This demonstrates that our proposed method is also capable of delivering comparable or favourable performance to Deep Learning alternatives.
>
> **Q2:** We thank the reviewer for pointing this out. We assume you are referring to the "KDD CUP 99" dataset. We considered including it, however this dataset has no meaningful temporal component as it is primarily used for anomaly/intrusion detection which is a supervised/unsupervised classification problem of independent samples for normal and abnormal states. We follow your request and introduce large-scale regimes within MNIST with 3000 samples/class with 5 regimes, for each sequence. In total, we collect 10 sequences with 15k samples each (15.000x784). We select $w=K_{\max}=100$, $\alpha=0.05$, $L=1000$, $p=2$.
>
> |AUC|COV|DD|FP|
> |---|---|---|---|
> |$0.92 \pm 0.07$|$0.95 \pm 0.05$|$30.9$ (20.5;520)|$0.8$ (0;2)|
>
> Additionally, we increase the number of change points to $10$ for the synthetic data, and shift a larger subset of 15/50 features.
> > Exponential
>
> |$\lambda$|AUC|||FP|||
> |---|---|---|---|---|---|---|
> ||$\tau=5$| $\tau=10$|$\tau=20$|$\tau=5$| $\tau=10$|$\tau=20$|
> |$0.25$|$1.0 \pm 0.0$|$1.0 \pm 0.0$|$1.0 \pm 0.0$|$0.0$ (0;0)|$0.0$ (0;0)|$0.0$ (0;0)|
> |$0.5$|$0.96 \pm 0.05$|$1.0 \pm 0.0$|$1.0 \pm 0.0$|$0.4$ (0;1)|$0.0$ (0;0)|$0.0$ (0;0)|
> |$0.75$|$0.42 \pm 0.1$|$0.98 \pm 0.05$|$1.0 \pm 0.0$|$5.8$ (4;7)|$0.2$ (0;1)|$0.0$ (0;0)|
>
> > Mixture (Gaussian/Exponential)
>
> |$\lambda / \sigma$|AUC|||FP|||
> |---|---|---|---|---|---|---|
> ||$\tau=5$| $\tau=10$|$\tau=20$|$\tau=5$| $\tau=10$|$\tau=20$|
> |$0.1$|$1.0 \pm 0.0$|$1.0 \pm 0.0$|$1.0 \pm 0.0$|$0.0$ (0;0)|$0.0$ (0;0)|$0.0$ (0;0)|
> |$0.5$|$0.96 \pm 0.05$|$1.0 \pm 0.0$|$1.0 \pm 0.0$|$0.4$ (0;1)|$0.0$ (0;0)|$0.0$ (0;0)|
> |$0.75$|$0.48 \pm 0.1$|$0.96 \pm 0.05$|$1.0 \pm 0.0$|$5.2$ (4;7)|$0.2$ (0;1)|$0.0$ (0;0)|
>
> References:
>
> [1] Hushchyn, M., Arzymatov, K., & Derkach, D. (2020). Online neural networks for change-point detection. arXiv preprint arXiv:2010.01388.

---

> ### Author Response · Authors · 2025-11-14
> **Answer (2/2)**
>
> Below, we answer the remaining questions.
>
> **Q3:** SWDCP is fully nonparametric. While random projections exhibit Gaussian tendencies due to the CLT, the Sliced Wasserstein Distance does not require any Gaussian assumptions. We already included a study about Gaussian tendencies of the random projections using the Shapiro-Wilk test across different dimensions $d$ with $N=100$ samples and  $L$ projection counts. Table 5 shows that normality becomes valid for $d>20$ and in most cases for $d>10$.
>
> We follow your request and now report the Mean Absolute Error (MAE) between the **theoretical quantiles** and **observed quantiles**. The theoretical quantiles are derived from a Gamma distribution based on the MoM estimates from the random projections involved in the calculation of the Sliced Wasserstein distance. The observed quantiles are calculated based on the empirical distribution of the random projections. For each dimension, we simulate two independent datastreams, each consisting of $d$ independent Gaussian distributions with a uniformly sampled mean. We vary $d$ and $L$, use fixed random seeds, and report the results for 10 trials below.
>
> | Dimension ($d$) | L=100 | L=1.000 | L=10.000 |
> | :---: | :---: | :---: | :---: |
> | 5 | $1.1226 \pm 0.1722$ | $0.3586 \pm 0.0326$ | $0.3751 \pm 0.0106$ |
> | 10 | $0.3479 \pm 0.0631$ | $0.3187 \pm 0.0638$ | $0.1499 \pm 0.0090$ |
> | 20 | $0.4014 \pm 0.0724$ | $0.1652 \pm 0.0373$ | $0.01347 \pm 0.0158$ |
> | 100 | $0.2912 \pm 0.0451$ | $0.1100 \pm 0.0154$ | $0.0470 \pm 0.0124$ |
> | 200 | $0.2987 \pm 0.0535$ | $0.1344 \pm 0.0485$ | $0.0448 \pm 0.0142$ |
>
>
>
> **Q4:** Theoretical runtime complexity as a function of  $L$, $d$, and $w$ is provided in Appendix C.2 Table 6. In addition, we include empirical runtime comparisons in Table 8, Table 9 against other OT-based methods to contextualize the efficiency of SWCPD.
>
> **Q5:** We now add a quantitative evaluation on the faithfulness of the "discriminative features" derived by our proposed method. We simulate a Mixture distribution with $d=50$ (25 Gaussian, 25 Exponential) with $T=500$ observations. We randomly select 10 features for which we inject a drift at $t>250$ with a drift magnitude uniformly sampled in $[-\delta,\delta]$. We let our method identify the 10 most discriminative features and mask the time series by removing the identified features. For validation, we use an independent oracle (KCPA), which has an AUC of 1.0 on the original data, and evaluate it on the masked data. Below, we report the average accuracy of removed features, and measure the AUC and the Covering change for the masked vs original data.
>
> |$\delta$ | 0.2 | 0.3 | 0.5 | 0.7 | 1.0 | 2.0 |
> |:------------|:----------------|:----------------|:----------------|:----------------|:----------------|:----------------|
> | **Accuracy** | $0.683 \pm 0.108$ | $0.767 \pm 0.075$ | $0.868 \pm 0.0745$ | $0.90 \pm 0.0816$ | $0.90 \pm 0.0816$ | $0.95 \pm 0.0764$ |
> | **AUC\_Drop** | $0.250 \pm 0.0$ | $0.250 \pm 0.0$ | $0.250 \pm 0.0$ | $0.250 \pm 0.0$ | $0.250 \pm 0.0$ | $0.25 \pm 0.0$ |
> | **COV\_Drop** | $0.490 \pm 0.012$ | $0.491 \pm 0.011$ | $0.499 \pm 0.002$ | $0.499 \pm 0.002$ | $0.50 \pm 0.00$ | $0.50 \pm 0.0$ |

---

### Official Review · Reviewer_6jLv · 2025-11-03

**Soundness:** 3
**Presentation:** 2
**Contribution:** 3
**Rating:** 6
**Confidence:** 5

**Summary:**

The paper proposes a novel method for explainable online change-point detection in high-dimensional data using the sliced Wasserstein distance. By transforming a multivariate time series into a one-dimensional signal, the method both detects change points and identifies which univariate series contribute most to the change. The detector uses a self-adaptive threshold, updating it for each window.

**Strengths:**

1. The paper tackles an interesting practical task: detecting change points and identifying which components of the time series contribute to those changes.

2. It provides theoretical guarantees for the distribution of the Sliced Wasserstein distance.

3. Visualizations and experiments on synthetic and real-world datasets demonstrate the method’s interpretability and change-detection ability, compared with multiple baselines.

4. The paper includes an ablation study with various parameter settings and reports runtime across different time-series lengths.

**Weaknesses:**

1. Although the paper targets high-dimensional data, the datasets in the experiments are relatively low-dimensional. The largest dimension is 50 in the change point task with synthetic datasets. It would be beneficial to include higher-dimensional settings in the explainability section as well. In addition, the runtime as a function of the dimension $d$ is not reported.
2. The main paper claims that the Sliced Wasserstein distance follows a Gamma distribution, but the theory is proved only for $p=2$.
3. In the synthetic-data explainability section, interpretability is shown only for Gaussian mean-shift cases. It would be more complete to include variance shifts and other distributions. In the synthetic change-point detection section, there is no baseline comparison.

Some minor comments:
1. Some notation appears before it is introduced. For example, $\delta$ in line 233, $K_{max}$ on line 249.
2. The notation of $p$ and $q$ is sometimes confusing and used interchangeably. For example, on lines 167 and 259. $p$ should denote the Wasserstein order, and $q$ the confidence level or quantile?

**Questions:**

1. In the experiments, what tolerance is used to terminate Algorithm 2 at line 3?
2. The number of change points is relatively small compared with the data dimension and the series length. What happens if you simulate many change points and shift a larger subset of series, for example, more than 2 change points across more than 3 series when $d=50$?
3. Is it challenging to provide a theoretical guarantee for change-point detection? For example, can the current theory be extended to quantify the uncertainty of correctly localizing the change point?
4. Is the dimensionality of the HAR dataset used in the experiments 250? What does “selected 25 instances” mean?

---

> ### Author Response · Authors · 2025-11-19
>
> We thank the reviewer for the thoughtful feedback and insightful comments.
>
> First, we want to comment on some weaknesses mentioned.
>
> **W1**: Theoretical runtime complexity as a function of  $L$, $d$, and $w$ is provided in Appendix C.2 Table 6.
>
> **W3**: We address this point in **Q5 Answer(2/2)** for reviewer rWAP, where we show the faithfulness of explanations for a Mixture distribution with $d=50$. This shows generalizability to non-Gaussian mean shifts.
>
> Thank you for commenting on the minor points in the notation. We will fix them for the revised manuscript.
>
> Below, we want to answer to your questions.
>
> **Q1**: For the synthetic experiments, we calculate one-shot contributions with Algorithm 1 for $q=0.95$, thus Algorithm 2 is terminated after a single iteration. This investigates the sensitivity of the contribution scores without further refinement, which is summarized in Table 1. It shows that those scores show stronger alignment to other explanation methods when the ratio of drifted feature to total features is smaller than $0.5$. For MNIST with pixels in $[0,1]$, we found that for two distinct digits, there are $245.08$ pixels on average, which show an absolute deviation above $0.1$. Changes below this are generally indistinguishable, such that this reduced set captures the most important pixels which are a valid representation of the original class, therefore we qualitatively terminated Algorithm 2 after 250 iterations.
>
> **Q2**: For this matter, we increase the number of change points to $10$ (before $2$) for the synthetic data, and shift a larger subset of 15/50 features (before $3$).
>
> > Exponential
>
> |$\lambda$|AUC|||FP|||
> |---|---|---|---|---|---|---|
> ||$\tau=5$| $\tau=10$|$\tau=20$|$\tau=5$| $\tau=10$|$\tau=20$|
> |$0.25$|$1.0 \pm 0.0$|$1.0 \pm 0.0$|$1.0 \pm 0.0$|$0.0$ (0;0)|$0.0$ (0;0)|$0.0$ (0;0)|
> |$0.5$|$0.96 \pm 0.05$|$1.0 \pm 0.0$|$1.0 \pm 0.0$|$0.4$ (0;1)|$0.0$ (0;0)|$0.0$ (0;0)|
> |$0.75$|$0.42 \pm 0.1$|$0.98 \pm 0.05$|$1.0 \pm 0.0$|$5.8$ (4;7)|$0.2$ (0;1)|$0.0$ (0;0)|
>
> > Mixture (Gaussian/Exponential)
>
> |$\lambda / \sigma$|AUC|||FP|||
> |---|---|---|---|---|---|---|
> ||$\tau=5$| $\tau=10$|$\tau=20$|$\tau=5$| $\tau=10$|$\tau=20$|
> |$0.1$|$1.0 \pm 0.0$|$1.0 \pm 0.0$|$1.0 \pm 0.0$|$0.0$ (0;0)|$0.0$ (0;0)|$0.0$ (0;0)|
> |$0.5$|$0.96 \pm 0.05$|$1.0 \pm 0.0$|$1.0 \pm 0.0$|$0.4$ (0;1)|$0.0$ (0;0)|$0.0$ (0;0)|
> |$0.75$|$0.48 \pm 0.1$|$0.96 \pm 0.05$|$1.0 \pm 0.0$|$5.2$ (4;7)|$0.2$ (0;1)|$0.0$ (0;0)|
>
> We hope this illustrates that our proposed method is also capable of handling a higher ratio of drifted features for shorter regimes.
>
> **Q3**: Our framework already provides statistical guarantees through hypothesis tests and p-values, which control false-alarm rates (uncertainty) by construction.
> Probably a fully Bayesian treatment of $\mathbb{P}(CP|S_{L}(\mathcal{D_{t}^{w}}))$ could be more elegant for quantifying uncertainties, but our overall aim is **nonparametric**, **high-dimensional**, **online**, **interpretable** CPD. A fully Bayesian generative model over high-dimensional distributions and SW slices would be intractable, unapplicable in real time, and therefore contradict our design principles.
>
> **Q4**: The HAR data we used was collected in [1], which has 12 features, and a total of 250 collections. We selected 25 representative instances described in Section C.2.3. There are several HAR datasets in the literature, for example [2], which has three dimensions, or [3], which has 561 dimensions. Additionally, we report the results averaged over the 30 time series of the HAR data from [3].
>
> |Method| AUC| Covering| FP| DD|
> |---|---|---|---|---|
> |SWCPD|$0.88 \pm 0.07$|$0.67 \pm 0.04$|$0.6$ (0;4)|$4.8$ (2.8;6.5)|
> |KCP|$0.85 \pm 0.06$|$0.82 \pm 0.07$|$2.5$ (0;8)|$3.7$ (1.0;7.7)|
> |e-divisive|$0.82 \pm 0.07$|$0.76 \pm 0.12$|$4.9$ (1;14)|$4.7$ (1.25;9.3)|
> |BOCPD|$0.76 \pm 0.06$|$0.53 \pm 0.09$|$0.1$ (0;1)|$2.8$ (1.8;4.2)|
> |OT-CPD|$0.73 \pm 0.06$|$0.52 \pm 0.07$|$0.15$ (0;1)|$1.8$ (0.5;4.16)|
>
> We hope this clarifies your questions and illustrates that our proposed method is tailored to high-dimensional data.
>
> References:
>
> [1] Ermshaus, Arik, et al. "Human activity segmentation challenge@ ECML/PKDD’23." International Workshop on Advanced Analytics and Learning on Temporal Data. Cham: Springer Nature Switzerland, 2023.
>
> [2] G. M. Weiss, K. Yoneda, and T. Hayajneh. Smartphone and Smartwatch-Based Biometrics Using Activities
> of Daily Living. IEEE Access, 7:133190–133202, 2019.
>
> [3] Reyes-Ortiz, Jorge, et al. "Human Activity Recognition Using Smartphones." UCI Machine Learning Repository, 2013, https://doi.org/10.24432/C54S4K.

---

### Author Response · Authors · 2025-11-28
**Rebuttal Revision**

We would like to thank all the reviewers again for their constructive feedback. In light of your comments and concerns, we have updated the manuscript accordingly.

In summary, we have

1) updated the **related work** with relevant literature pointed out (EQ8h,7mTU)
2) updated the **stopping criterion** for Algorithm 2 (7mTU,6jLv)
3) included an empirical study on the **validity/faithfulness** of the discriminative features obtained using Algorithm 2 (rWAP,6jLv)
4) included an **additional high-dimensional dataset** (7mTU,6jLv)
5) extended the change point detection study by **three baseline methods** (RuLIFS, and two deep-learning approaches) (EQ8h,rWAP)

We hope this clarifies your comments and concerns sufficiently. Please let us know if there are any open concerns.

---

### Meta-Review · Area_Chair_qKkG · 2026-01-06

**Summary:**

Reviewer 6jLv had some concerns about experimental settings.

Reviewer rWAP is concerned that the samples within a window will not be IID which may affect the method.  Specifically, the claim in the paper "Despite temporal correlations, the i.i.d. nature of the random projections ensures the validity of our statistical bounds." is not substantiated. Random projections of a collection of dependent random vairables will be dependent in general.

Reviewer 7mTU seems to be generally supportive, however, in my opinion, their review has enough criticism to also justify a rejection score. Interestingly, they also pointed out to the issue above about assuming away IIDnes in windows.

Reviewer EQ8h was only concerned with missing baselines.

**Reviewer Concerns:**

The authors add some new experimental results to address Reviewer 6jLv's concerns in a table. But I am not sure if they run high-dimensional experiments.

The authors mention both in their rebuttal and in the paper that the random projections take care of dependencies within a window. This is not clear to me (apparently also to the reviewer). Another point: They said "We don't rely on i.i.d. samples within the windows". But in the paper they explicitly mention this is leveraged in their statement "Despite temporal correlations, the i.i.d. nature of the random projections ensures the validity of our statistical bounds."

The authors' reply to Reviewer 7mTU was, instead of providing a theoretical rigorous reply, to provide empirical results.

**Reviewer Scores:**

I do not believe the rebuttal would have changed most of the reviewers' scores. Reviewer EQ8h is the only exception having stated they would raise their score to 6. This is expected since their only substantial comment was lack of baselines which the authors have addressed.

---

### Decision · Program_Chairs · 2026-01-26

Reject